# Inhibition of host NOX1 blocks tumor growth and enhances checkpoint inhibitor–based immunotherapy

Jimmy Stalin[1,2] , Sarah Garrido-Urbani[1], Freddy Heitz[3], Cédric Szyndralewiez[3], Stephane Jemelin[1], Oriana Coquoz[2], Curzio Ruegg[2,*] , Beat A Imhof[1,4,*]

NADPH oxidases catalyze the production of reactive oxygen species and are involved in physio/pathological processes. NOX1 is highly expressed in colon cancer and promotes tumor growth. To investigate the efficacy of NOX1 inhibition as an anticancer strategy, tumors were grown in immunocompetent, immunodeficient, or NOX1-deficient mice and treated with the novel NOX1-selective inhibitor GKT771. GKT771 reduced tumor growth, lymph/angiogenesis, recruited proinflammatory macrophages, and natural killer T lymphocytes to the tumor microenvironment. GKT771 treatment was ineffective in immunodeficient mice bearing tumors regardless of their NOX-expressing status. Genetic ablation of host NOX1 also suppressed tumor growth. Combined treatment with the checkpoint inhibitor anti-PD1 antibody had a greater inhibitory effect on colon carcinoma growth than each compound alone. In conclusion, GKT771 suppressed tumor growth by inhibiting angiogenesis and enhancing the recruitment of immune cells. The antitumor activity of GKT771 requires an intact immune system and enhances anti-PD1 antibody activity. Based on these results, we propose blocking of NOX1 by GKT771 as a potential novel therapeutic strategy to treat colorectal cancer, particularly in combination with checkpoint inhibition.

## Introduction

Colorectal carcinoma is the second leading cause of cancer-related mortality in developed countries (1). Surgical resection is currently the treatment of choice. However, ~30% of node-positive patients develop local recurrence or distant metastasis within 5 yr of surgery and die of the disease (2). Dysregulated expression of proinflammatory cytokines and growth factors contributes to the development of colorectal tumors and tumor progression by stimulating tumor angiogenesis and recruiting tumor-promoting immune cells. The release of proinflammatory cytokines in response to surgery further promotes tumor progression (3). Tumor angiogenesis, that is,

the de novo formation of tumor-associated vessels, is crucial for tumor progression, whereas in the absence of angiogenesis, tumors remain dormant as microscopic dormant lesions that can persist for years (4). In addition to tumor cells, stromal cells and immune cells, including bone marrow–derived monocytes can induce angiogenesis through a process called angiogenic switch. This is the result of an imbalance in the production of pro- versus anti-angiogenic factors, eventually leading to the sprouting of activated endothelial cells from the preexisting, quiescent vasculature (5, 6). Many angiogenic factors (e.g., VEGF and FGF) and their receptors (e.g., VEGFR-2 and FGF-Rs) have been identified as therapeutic targets, and inhibitors of these molecules (e.g., bevacizumab and sunitinib) are currently in clinical use or under development as novel anti-angiogenic agents to suppress cancer progression (7).

NADPH oxidases (NOXs) catalyze the production of reactive oxygen species (ROS). ROS are involved in different physiological and pathological processes, including cancer, and their effect depends on concentration and cellular localization (8). The NOX family of enzymes, which comprises seven isoforms (NOX1, NOX2, NOX3, NOX4, NOX5, DUOX1, and DUOX2), transports electrons across the cell membrane during the production of superoxide through the reduction of oxygen (9). NOX enzymes play a major role in numerous cellular processes such as apoptosis, host defense against pathogens, intracellular signal transduction, and angiogenesis (10). NOX1, NOX2, and NOX4 expression in cancer cells promotes tumor growth and metastasis in several cancers, including melanoma, gastric, pancreatic, and colon tumors (11). The NOX1 isoform is up-regulated in colon cancer (12), and its overexpression correlates with inflammation rather than tumorigenesis (13, 14). NOX1 is highly expressed in colon cancer cell lines and promotes proliferation (15). Small hairpin RNA-mediated NOX1 silencing suppresses tumor growth in mouse models of colon cancer, and inhibition of NOX activity with pharmacological pan-NOX inhibitors decreases cancer cell proliferation without inducing apoptosis (16, 17).

NOX1 is expressed in epithelial cells, pericytes, endothelial cells, vascular smooth muscle cells, and immune cells (18, 19, 20, 21).

---

[1]Department of Pathology and Immunology, Medical Faculty, University of Geneva, Geneva, Switzerland   [2]Department of Oncology, Microbiology and Immunology, Faculty of Science and Medicine, University of Fribourg, Fribourg, Switzerland   [3]Genkyotex S.A Forum 2, Archamps Technopole, Saint-Julien-en-Genevois, France   [4]Medicity Research Laboratory, University of Turku, Turku, Finland

Correspondence: jimmy.stalin@unifr.ch; curzio.ruegg@unifr.ch; beat.imhof@unige.ch
*Curzio Ruegg and Beat A Imhof contributed equally to this work

However, the role of NOX1 in tumor-associated immune cells remains to be fully characterized. NOX1/2 KO mice show an enhanced proinflammatory macrophage signature and increased frequency of M1 proinflammatory macrophages in tumors growing in these mice (22). Whether this effect is mediated directly and exclusively by NOX1 remains unclear. Furthermore, in the aortic sinus of diabetic ApoE−/− mice, NOX1-derived ROS promote macrophage accumulation and inflammation, suggesting that NOX1 modulates macrophage recruitment and may contribute to vascular pathologies (23).

NOX1 is involved in immune-related disorders or immune cell regulation. NOX1 is up-regulated in blood vessels in an in vivo model of hypertension and is overexpressed in the atherosclerotic plaque of patients with cardiovascular diseases or with established diabetes mellitus (24). These reports are consistent with the observations that combined inhibition of NOX1 and NOX4 with pharmacological inhibitors in mice leads to dose-dependent atheroprotection (25). Taken together, these findings suggest that NOX1 is a promising therapeutic target for the management of immune/inflammatory events in cancer and vascular pathologies.

Here, we show that GKT771, a novel, potent, and highly selective pharmacological inhibitor of NOX1, or genetic deletion of NOX1 in mice reduced tumor growth in preclinical models of colorectal cancer and melanoma in immunocompetent mice. NOX1 inhibition decreased tumor angiogenesis and lymphangiogenesis and modulated the composition of tumor-associated immune cells in colorectal cancer by promoting the recruitment of immune/inflammatory cells consistent with the observed decrease in tumor growth. The immunostimulatory function of GKT771 was essential for its antitumor activity and combination treatment with GKT771, and anti-PD1 antibody showed enhanced inhibition of tumor growth.

# Results

## GKT771 inhibits tumor growth, angiogenesis, and lymphangiogenesis in MC38-derived colon carcinoma

We and others previously showed that broad-spectrum NOX inhibitors targeting several NOX isoforms decrease the growth of experimental tumors (16, 17). However, because of the lack of specificity of the currently available inhibitors, the contribution of each individual NOX isoform to tumor growth and progression remains undetermined. A potent and highly selective NOX1 inhibitor (GKT771) has been developed by GenKyoTex. As shown in Fig S1A, GKT771 inhibits NOX1 with an inhibitory constant Ki of 60 ± 6 nM. It is highly selective over NOX4 (Ki = 4,000 ± 400 nM) and is inactive against all other NOX isoforms. Moreover, GKT771 shows to be inactive on all counterscreen assays, including xanthine oxidase, glucose oxidase, and scavenging assays (data not shown). Together with the availability of NOX1-deficient mice, GKT771 provides a unique opportunity to investigate selective NOX1 inhibition as a potential anticancer strategy. To this end, we used two syngeneic tumor models based on MC38 colon carcinoma and B16-F10 mouse melanoma subcutaneously transplanted into immunocompetent C57/BL6 animals. In these two models, NOX1 and NOX4 expression was different among the cell lines and conditions: NOX1 expression was higher in B16F10 cells both in vitro and in vivo, whereas NOX4 expression was higher in MC38

cells in vitro but similar to B16F10 in vivo (Fig S1A and 1B). When tumor size reached approximately 50 mm³, mice were randomly assigned to five groups and treated by oral gavage with the NOX1-specific inhibitor GKT771, the broad-spectrum inhibitor GKT831, or the anti–VEGFR-2 blocking antibody DC101. Treatment with the specific inhibitor GKT771 suppressed MC38 and B16-F10 tumor growth compared with vehicle (VL) alone (Figs 1A and B, and S1C). Treatment with the previously available NOX1/NOX4 inhibitor GKT831 also suppressed B16-F10 tumor growth, whereas it had only a weak effect on MC38 tumors (Figs 1A and 1B, and S1D). Anti-angiogenic treatment with the DC101 antibody also decreased tumor growth (Fig 1A).

Because NOXs regulate tumor angiogenesis (13, 15, 24), the number of blood vascular and lymphatic endothelial cells was quantified in tumor cell suspensions by flow cytometry in the CD45-negative cell fraction using antibodies against CD31 and GP38 (Fig S1E). The GKT771 inhibitor and DC101 antibody reduced the number of CD45⁻/CD31⁺/GP38⁻ blood vascular and CD45⁻/CD31⁺/GP38⁺ lymphatic endothelial cells (Fig 1C and D). Combined GKT771/DC101 treatment showed marginal additive effects over treatment with GKT771 alone.

## GKT771 inhibits endothelial cell proliferation and angiogenesis directly and indirectly

Angiogenesis of tumor blood or lymphatic vessels can be inhibited directly by affecting endothelial cells or indirectly through the modulation of pro-angiogenic factors produced by tumor cells or cells of the microenvironment. To determine whether NOX1 inhibition acts via direct or indirect pathways, the effect of GKT771 was first tested using an ex vivo aortic ring angiogenesis assay. Both NOX1 and dual NOX1/4 inhibitors significantly decreased the branching area of sprouting vessels, a surrogate parameter for angiogenesis (Fig 1E). This result is consistent with a direct effect on endothelial cells. To test whether GKT771 also inhibits angiogenesis in vivo, a Matrigel plug assay was performed by injecting bFGF-loaded Matrigel subcutaneously. At 1 wk after injection, de novo blood vessels were detected by in vivo staining of blood vessels with FITC-lectin and monitoring fluorescence in plug sections. Treatment with GKT771 significantly decreased the vascularization of plugs (Fig 1F).

To examine the anti-angiogenic mechanism of GKT771, its potential direct effect on endothelial cell proliferation was tested. Assessment of BrdU incorporation in cultured mouse primary lung microvascular endothelial cells (mMVEC-L) treated with GKT771 showed significant inhibition of cell growth in a concentration-dependent manner (Fig 1G). To determine whether NOX inhibitors could block angiogenesis indirectly by decreasing the production of pro-angiogenic factors, we measured VEGF-A and PLGF (Placental Growth Factor) levels by ELISA in the plasma of tumor-bearing mice and in the supernatant of cancer cells treated in vitro with NOX inhibitors. The results showed that both GKT771 and GKT831 decreased the plasma concentration of VEGF-A and PLGF in MC38 and B16-F10 tumor-bearing mice (Figs 1H and S1F). NOX inhibitors also decreased the secretion of VEGF-A and PLGF by MC38 and B16-F10 cancer cells in vitro (Fig 1H). DC101 treatment in vivo increased the levels of both vascular growth factors, consistent with a positive adaptive response (Fig S1G). Treatment with GKT771 also decreased the concentration of the lymphangiogenic factor VEGF-C in the plasma of tumor-bearing mice (Fig 1I). GKT771 and GKT831 significantly

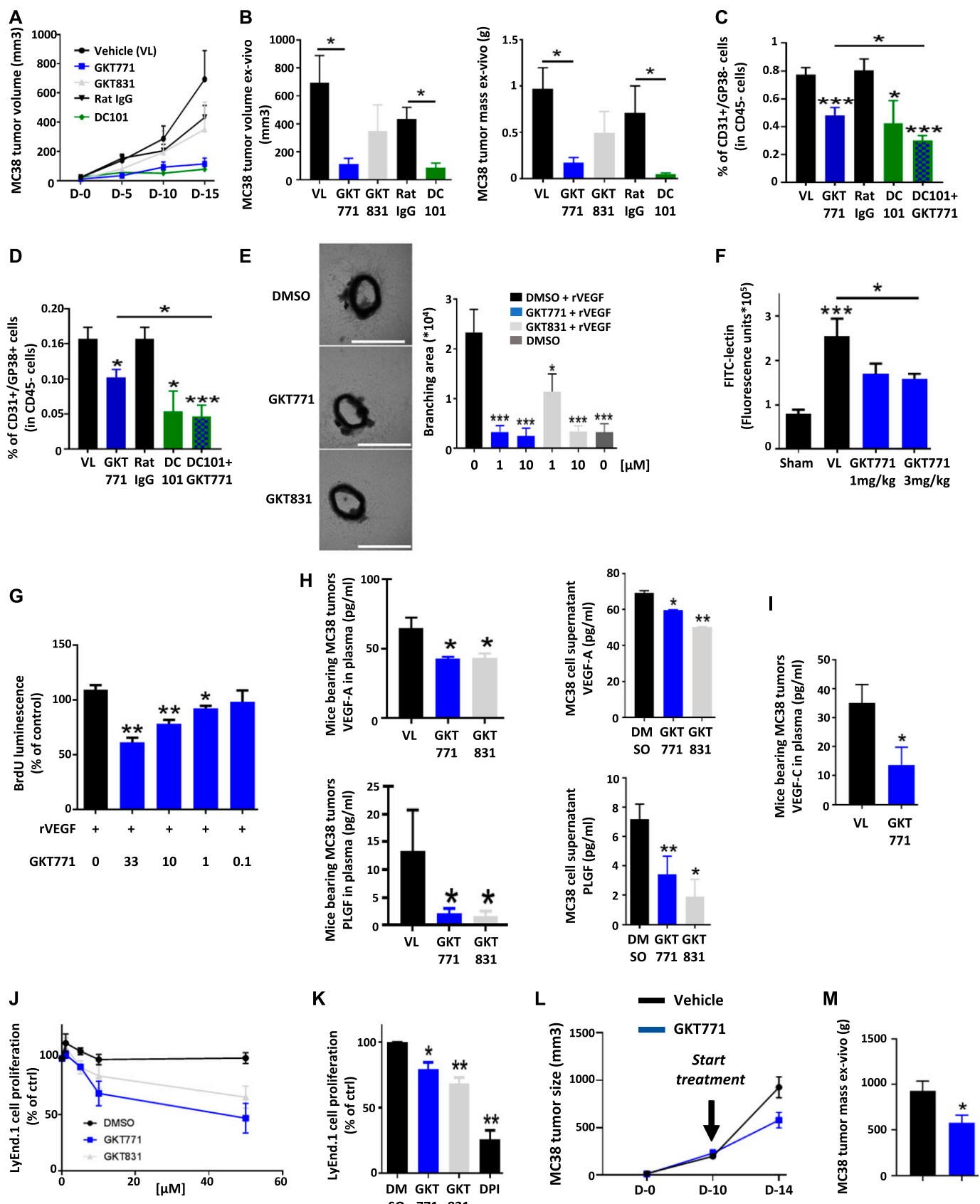

inhibited lymphatic endothelial cell growth in a dose-dependent manner (Fig 1J) and significantly inhibited cell proliferation by approximately 20% at 10 μM (Fig 1K). The wide-spectrum NOX/DUOX inhibitor DPI served as a positive control. The inhibitory effects correlated with an increased number of dead cells (Fig S1H). However, analysis of VEGF-A mRNA expression in cancer cells treated in vitro with GKT771 or GKT831 showed no decrease in expression. In contrast, VEGF-A mRNA measurement in tumors treated in vivo with GKT771 showed a trend towards decreased levels (data not shown). This suggests that the decrease of VEGF-A in supernatant and mouse plasma is related to a decreased cell proliferation in vitro and a decreased tumor growth in vivo.

Taken together, these results indicate that the specific NOX1 inhibitor GKT771 suppressed tumor growth and blocked angiogenesis and lymphangiogenesis in melanoma and colorectal syngeneic cancer models. These effects were mediated by decreased production of angiogenic factors by cancer cells and direct inhibition of endothelial cell proliferation and sprouting.

## GKT771 inhibits growth of MC38 colon carcinoma and B16F10 melanoma tumors in a clinically relevant model

Starting treatment at 50 mm$^3$ tumor size may overestimate the potential effects on larger tumors. We, therefore, repeated experiments with GKT771 by starting treatment at tumor sizes of around 200 mm$^3$, for both MC38 and B16F10 tumor models, which can be considered more representative of tumors in clinical settings. Mice were treated during 4 d compared with 10 d in the previous protocol. Under these new experimental conditions, the 4-d GKT771 treatments inhibited tumor growth (measured as tumor volume and tumor weight at the end of the experiment) by 35% for MC38 (Fig 1L and M) and 50% for B16F10 (Fig S1I and J).

These results indicate that GKT771 is efficient in inhibiting growth of well-established tumors, as it is the case in clinical situations.

## NOX1-deficient mice show reduced angiogenesis and do not respond to GKT771

To determine the involvement of NOX1 endogenously expressed by host cells in promoting tumor growth, B16-F10 and MC38 cancer cells were transplanted into wild-type (WT) or NOX1-deficient mice,

followed by treatment with GKT771. Both tumors grew at a slower rate in NOX1 KO mice compared with WT mice (Figs 2A and S2A). GKT771 treatment had no additional antitumor effects in NOX1 KO mice (Figs 2B and S2B).

To confirm the antitumor properties of GKT771, the direct effect of the compounds on tumor cells was tested in vitro. The two inhibitors decreased the proliferation of MC38 colon and B16-F10 melanoma cancer cells without affecting cell viability (data not shown). This was consistent with previous reports showing that inhibition of NOXs leads to decreased cancer cell proliferation without inducing apoptosis (17). These results confirmed that GKT771 is a specific inhibitor of host NOX1 and displays minor effects on cancer cell viability.

By contrast, the anti-VEGFR-2 antibody DC101 decreased tumor growth, and the effect was significantly stronger in NOX1-deficient mice than in WT mice (Fig 2C and D). This suggests that VEGFR-2 and NOX1 promote tumor growth through different mechanisms of action. Quantification of endothelial cells in MC38 tumors in NOX1-deficient mice showed significantly lower numbers of CD31$^+$GP38$^-$ blood vascular and CD31$^+$GP38$^+$ lymphatic endothelial cells in NOX1-deficient mice compared with WT mice (Fig 2E).

To directly test the angiogenic potential of host NOX1, an ex vivo aortic ring assay was performed by comparing the branching area of rings derived from WT and NOX1-deficient mice treated in vitro with GKT771 or vehicle only (control DMSO). The results showed that the branching area of aortic rings was smaller in NOX1-deficient mice compared with WT mice and GKT771 inhibited the branching area of aortic rings in WT mice, whereas it had no effect on NOX1-deficient mice (Fig 2F).

To determine whether the reduced angiogenic potential in NOX1-deficient mice was caused by a general decrease in the number of endothelial cells or a specific angiogenic deficit, we first determined the number of endothelial cells in lungs and MC38 tumors in WT and NOX1 KO mice by flow cytometry. The numbers of blood vascular (CD45$^-$/CD31$^+$/GP38$^-$) and lymphatic endothelial (CD45$^-$/CD31$^+$/GP38$^+$) cells were significantly lower in tumors from NOX1-deficient mice than in those from WT mice. Such a difference was not observed in the lungs of the same animals (Figs 2G and H, and S2C and D). The number of CD45$^+$/CD11b$^+$/F4-80$^+$ activated macrophages in lungs and tumors of NOX1-deficient and WT mice remained unchanged (Fig 2I).

---

**Figure 1. GKT771 inhibits MC38 colon carcinoma tumor growth by targeting angiogenesis and lymphangiogenesis.**
Mice with established MC38 tumors were treated with GKT771 (NOX1 inhibitor, n = 12), GKT831 (NOX1/4 inhibitor, n = 12), DC101 (anti–VEGFR-2 antibody, n = 6), and the corresponding IgG (n = 9) or vehicle (VI) (n = 14). **(A)** Tumor size was measured in vivo using a caliper (D-0 to D-15) every 5 d. **(B)** Tumor size and mass were measured ex vivo at the end of the experiment at day 15. **(C, D)** Tumors from each mouse were isolated, and blood vascular endothelial cells (CD45$^-$/CD31$^+$/GP38$^-$) and lymphatic endothelial cells (CD45$^-$/CD31$^+$/GP38$^+$) were analyzed by flow cytometry. The percentages of blood vascular cells (C) and lymphatic endothelial cells (D) in the differently treated groups are shown. **(E)** Aortic rings from C57/BL6 mice were stimulated with mouse rVEGF-A in the presence of GKT771 or GKT831 (1 or 10 μM) and compared with the vehicle DMSO-treated group as the negative control (n = 7) (0). Images show the rings with vascular branches and the graph shows the results of quantification of the branching area (right). **(F)** Vascularization in angioreactors as assessed by the FITC-lectin detection system. Mice implanted with angioreactors premixed with (VL, GKT771) or without (Sham) angiogenic-modulating factors were treated with vehicle (Sham, VL) or GKT771 at 1 and 3 mg/kg/d for 15 d (n = 12). **(G)** Inhibition of cell proliferation in VEGF-treated mMVEC-L cells. Concentration-dependent inhibition of cell proliferation by GKT771 in mMVEC-L cells treated with VEGF (20 ng/ml) for 24 h. Data represent the results from three experiments expressed as the mean ± SD of percentages of the controls (without VEGF treatment). Statistical analysis was performed using the unpaired t test *P < 0.05. Scale bars were added in the aortic ring images by using Image J software. White scale bar have a size of 1,000 μm. **(H)** Plasma of mice bearing MC38 tumors and the supernatants of MC38 cancer cells in culture were collected after GKT771, GKT981, or vehicle treatment and VEGF-A or PLGF was quantified by ELISA. **(I)** Quantification of VEGF-C in the plasma of mice bearing MC38 tumors after treatment with GKT771 or vehicle. **(J)** Proliferation of lymphatic endothelial LyEnd.1 cells in the presence of graded concentrations of GKT771 or GKT981. **(K)** Inhibition of LyEnd.1 cell proliferation in the presence of 10 μM GKT771, 10 μM GKT831, or 10 μM of the broad-spectrum NOX inhibitor DPI. Mice with established MC38 tumors were treated at 200 mm$^3$ with GKT771 (NOX1 inhibitor, n = 10) vehicle (VI) (n = 10). **(L, M)** Tumor size were measured in vivo using a caliper (D-0 to D-15) (L), and tumor mass was measured ex vivo at the end of the experiment at day 15 (M).

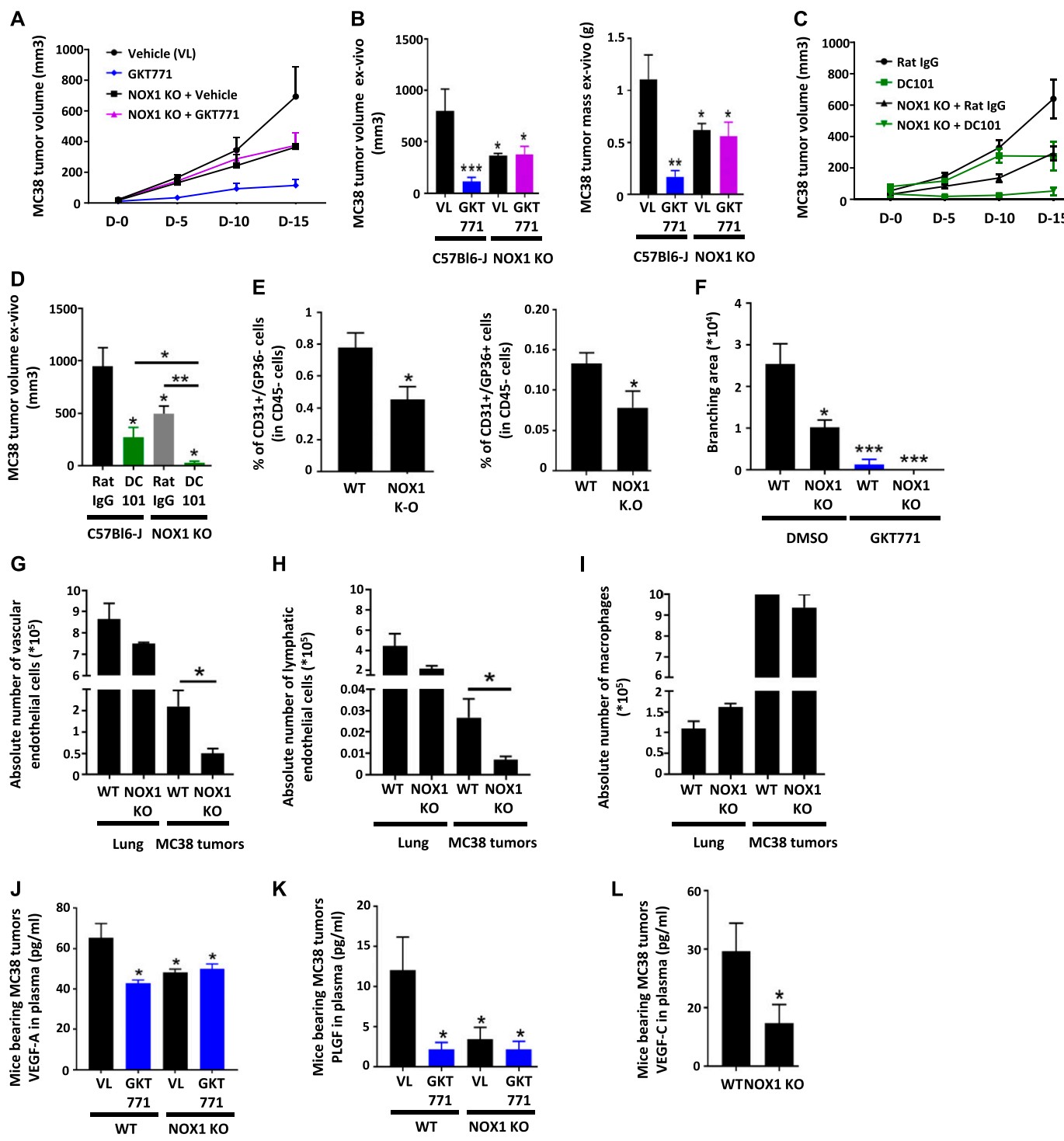

**Figure 2. NOX1-deficient mice do not respond to GKT771 and exhibit reduced tumor growth, angiogenesis, and lymphangiogenesis.**
C57/BL6 and NOX1-deficient mice with established MC38 tumors were treated with GKT771 (n = 11), DC101 (n = 6), rat IgG (n = 6), or vehicle (n = 13). **(A, B)** Tumor size was measured in vivo using a caliper (D-0 to D-15) every 5 d. **(C, D)** Tumor size and mass were measured ex vivo at the end of the experiment. **(E)** Tumors from C57/BL6 and NOX1-deficient mice were isolated, and blood vascular endothelial cells (CD45⁻/CD31⁺/GP38⁻) and lymphatic endothelial cells (CD45⁻/CD31⁺/GP38⁺) were analyzed by flow cytometry (n = 13 and n = 11, respectively). The percentages of blood vascular cells and lymphatic endothelial cells in WT and KO mice are shown. **(F)** Aortic rings from WT and NOX1-deficient mice were stimulated with mouse rVEGF-A and exposed to GKT771 or DMSO as the negative control (n = 7). Branching areas were quantified and graphed. **(G–I)** Absolute numbers of blood endothelial cells (CD45⁻/CD31⁺/GP38⁻) (G), lymphatic endothelial cells (CD45⁻/CD31⁺/GP38⁺), (H) and macrophages (CD45⁺/CD11b⁺/F4-80⁺) in lungs and tumors (I) were determined after sorting by cytometry from WT or KO mice (n = 4). **(J–L)** Plasma of WT and KO mice bearing MC38 tumors treated with GKT771 or DMSO was used to quantify VEGF-A (J), PLGF (K), and VEGF-C (L) by ELISA.

The angiogenic components of the host environment were further investigated by determining the plasma concentrations of the angiogenic factors VEGF-A and PLGF in NOX1-deficient versus WT mice bearing MC38 or B16F-10 tumors. The results showed that the concentrations of both factors were lower in NOX1 KO than in WT tumor mice. Treatment with GKT771 had no further effect in NOX1-deficient animals (Figs 2J and K, and S2E). Plasma VEGF-C was also decreased in NOX1-deficient versus WT MC38 tumor-bearing mice (Fig 2L).

Taken together, these results suggest that host NOX1 contributes to tumor growth, angiogenesis, and lymphangiogenesis. NOX1 deficiency affects tumor angiogenesis but not developmental angiogenesis and macrophages in healthy organs. Therefore, host NOX1 emerges as an attractive target for antitumor treatment.

### Host macrophages are not a main target for GKT771 antitumor effects

The macrophage signatures in NOX1/2 double KO mouse tumors differ from those in WT mice (22). Although no changes in total macrophage counts were observed in NOX1 KO animals, considering their angiogenic potential, we nevertheless investigated whether GKT771 treatment affected the phenotype of macrophages in tumors. FACS analysis of CD45$^+$ cells harvested from dissociated MC38

colorectal tumors detected a slight but significant increase of MHC class II$^+$/Ly6C$^+$, type I inflammatory macrophages (Fig 3A). DC101 treatment substantially increased the number of inflammatory macrophages in tumors, which nearly doubled in response to combination treatment with GKT771. However, GKT771 or DC101 alone or in combination did not affect the ratio of mature (CD45$^+$/CD11b$^+$/F4-80$^+$) and immature (CD45$^+$/CD11b$^+$/F4-80$^-$) macrophages or the frequency of total–tumor-associated macrophage (TAM) populations (CD45$^+$/CD11b$^+$/CD68$^+$/F4-80$^+$ and CD45$^+$/CD11b$^+$/CD68$^+$/F4-80$^-$) (Fig 3B and C).

To determine whether inflammatory macrophages are essential for the antitumor effect of GKT771, the macrophage population was depleted in C57Bl6/J mice using clodronate liposomes (26). Clodronate treatment started before the grafting of MC38 carcinoma cells and continued through the entire experimental period. When tumors reached 50 mm$^3$ in volume, the mice were treated with GKT771 or vehicle alone. The results showed that depletion of macrophages had no effect on the antitumor activity of GKT771 (Fig 3D–F).

These results indicate that NOX1 inhibition increases the number of MHC class II$^+$/Ly6C$^+$ proinflammatory macrophages in tumors, but this increase is not essential for the antitumor effect of GKT771.

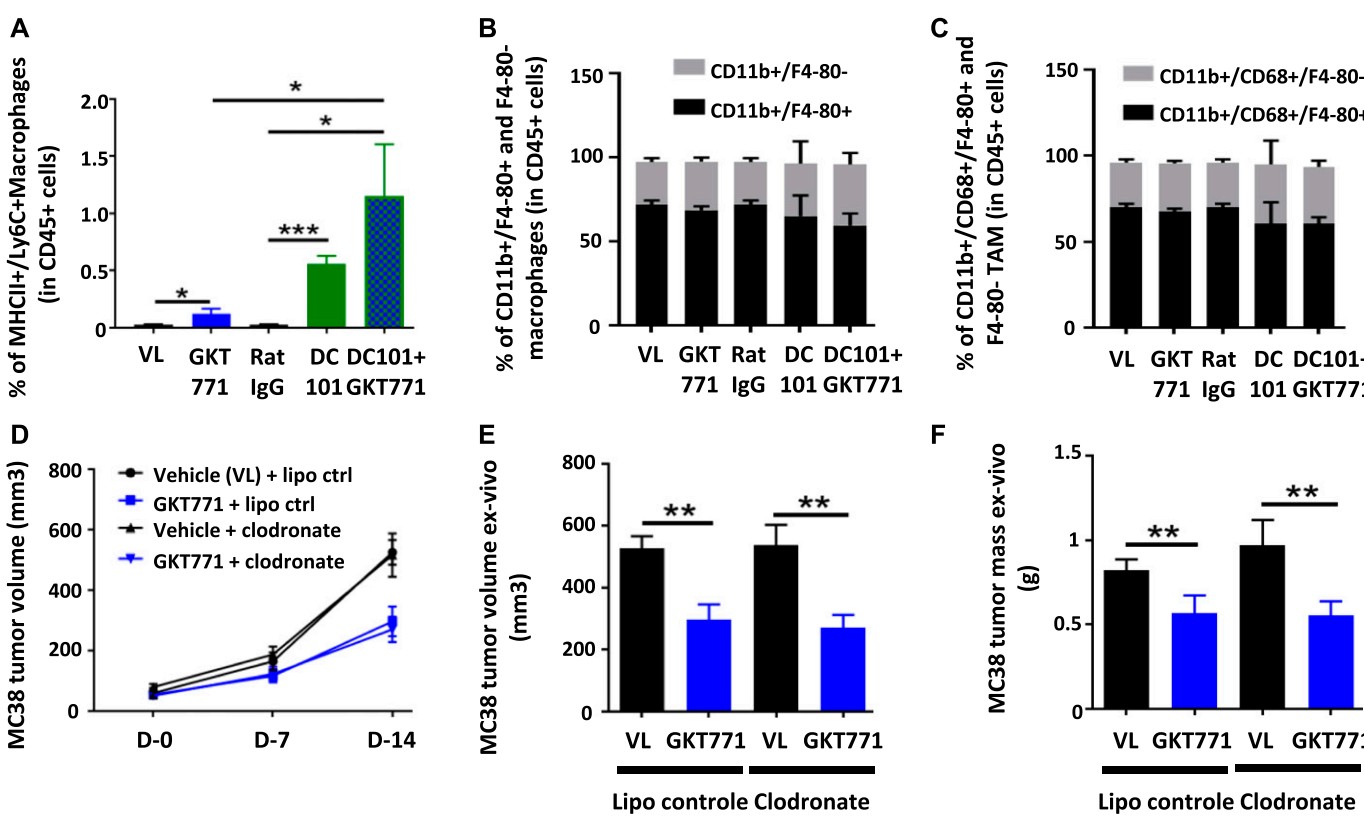

**Figure 3. GKT771 increases proinflammatory macrophages in tumors, although they are not essential for its antitumor effect.**
**(A–C)** MC38 tumors from C57/BL6 mice were excised and proinflammatory TAM (CD45$^+$/MHC2$^+$/Ly6C$^+$), total macrophages (CD45$^+$/CD11b$^+$/F4/80$^+$), and TAM (CD45$^+$/CD68$^+$/F4/80$^+$) were stained with the corresponding antibodies and analyzed by flow cytometry (n = 11 for the vehicle, GKT771, and rat IgG groups and n = 6 for DC101 and DC101+GKT771 groups). The percentages of proinflammatory TAM (A), total macrophages (B), and TAM (C) from the differently treated groups of animals are shown. **(D)** C57/BL6 mice were treated with clodronate liposomes before tumor cell engraftment. When tumors were established, the mice were treated with GKT771 (n = 10 mice per group) and the size of tumors was measured with a caliper. **(E, F)** Tumor size and mass were measured ex vivo at the end of the experiment.

### The immune system contributes to the antitumor effect of GKT771

To determine whether immune system cells other than macrophages contribute to the antitumor effect of GKT771, MC38 tumor growth was examined in immunodeficient NOD SCID Gamma (NSG) mice. These mice lack T and B lymphocytes and NK cells. GKT771 treatment had a minor and insignificant antitumor effect in these animals (Fig 4A and B). To confirm that the antitumor effect of GKT771 was not directly dependent on NOX1 expression in cancer cells, we studied the effect of GKT771 on the growth of DLD1 and LoVo cells, two human colon cancer cell lines expressing different levels of NOX1 (Fig S3A), grafted onto NSG mice. Although GKT771 treatment did not decrease tumor growth in these immunodeficient mice (Fig 4C–F), GKT771 suppressed cancer cell growth in vitro (Fig S3B and C) without inducing cancer cell death (Fig S3D). As shown for MC38 cells, the effect of GKT771 on DLD1 cell growth was dose dependent (Fig S3E).

GKT771 treatment decreased the percentage of vascular endothelial cells in DLD1 tumors (Fig 4G), whereas it had no effect on lymphatic endothelial cells and macrophages (Fig 4H and I). The total number of endothelial cells in the healthy lungs of NSG tumor-bearing mice was comparable with that in immunocompetent mice (Fig 4J–L).

Taken together, these results indicate that the host immune system contributes substantially to the antitumor effect of GKT771, whereas effects on angiogenesis and cancer cells do not or only marginally contribute to decreasing tumor growth in these models.

### GKT771 tumor blocking effects depend, at least in part, on interferon γ activity

The cytokine IFN-γ produced by activated T lymphocytes is essential for stimulating antitumor immunity (27, 28). We, therefore, investigated the effect of blocking IFN-γ on the antitumor effect of GKT771. Immunocompetent, MC38 tumor-bearing mice were treated with a blocking anti–IFN-γ antibody or GKT771 alone and in combination. Analysis of the kinetics of MC38 tumor growth and tumor size showed that anti–IFN-γ antibody treatment increased tumor growth and partially reduced the antitumor effect of GKT771 (Fig 5A and B). Similar results were obtained using NOX1 KO mice treated with anti–IFN-γ antibody (Fig 5C).

Taken together, these results demonstrate that the effect of the NOX1 inhibitor GKT771 on decreasing tumor growth is, at least in part, dependent on IFN-γ.

### GKT771 enhances the antitumor activity of the immune checkpoint inhibitor anti–PD-1 antibody

Tumors possess the ability to turn off antitumor immunity by decreasing the recruitment and activation of tumor-infiltrating T-lymphocytes (TILs) and by changing their phenotype (29). Moreover, anti-angiogenic drugs modulate tumor endothelial cells by allowing differential infiltration and activation of immune cells, thereby promoting antitumor immunity (30, 31). As NOX1 inhibition increased tumor infiltration by inflammatory macrophages, we tested the effect of GKT771 on TILs and tumor-associated natural killer T cells (NKT) cells, two cell types with potent antitumor

activities (32, 33). FACS analysis of the effects of GKT771, DC101, and combination thereof on immune cells infiltrating MC38 tumors, showed that GKT771 treatment did not change the percentages of CD4$^+$ and CD8$^+$ T cells or B lymphocytes, whereas DC101 antibody treatment decreased CD4$^+$ and increased CD8$^+$ T cells (Fig 6A–C). Both compounds increased the percentage of NKT cells when used as single agents, whereas in combination, they had no additive effects (Fig 6D).

Checkpoint blockade anti-PD1, anti–PD-L1, and anti-CTLA/4 antibodies promote both CD4$^+$ and CD8$^+$ T cell activation (34). However, their therapeutic efficacy is often limited, especially when used alone (35). Their antitumor activity in vivo increases when used in combination with drugs targeting tumor angiogenesis or other pathways (30, 31, 36). We, therefore, combined GKT771 tumor treatment with anti-PD1 antibody. This combination blocked MC38 tumor growth more efficiently than GKT771 or anti–PD-1 alone, and two animals out of seven were rendered completely tumor free (Fig 6E and F).

To determine whether lymphocytes were responsible for the tumor blocking effect, populations of TILs were analyzed in combined and single drug-treated animals. Compared with GKT771, anti–PD-1 antibody greatly increased the CD8$^+$ T lymphocyte fraction within TILs, resulting in a relative decrease in the numbers of CD4$^+$ T lymphocytes. This effect was maintained when anti–PD-1 antibody was combined with GKT771 (Fig 6G and H). In B lymphocytes, no relevant differences were detected in response to monotherapy, whereas combination treatment drastically increased the number of these cells (Fig 6I). Both compounds increased NKT cells when used alone, and the combined treatment potentiated this effect (Fig 6J). The same potentiating effect was observed in inflammatory macrophages (Fig 6K) despite the effect of PD-1 antibody treatment on decreasing TAM percentages in tumors (Fig S4A and B). The potentiating effect of host NOX1 inhibition and anti–PD-1 antibody treatment on tumor growth was confirmed in NOX1-deficient mice treated with anti–PD-1 antibody (Fig 6L).

In conclusion, combination treatment with a NOX1 inhibitor and anti-PD1 antibody was more effective for blocking tumor growth than single agent therapy, and the effect was confirmed in NOX1-deficient mice. Combined treatment was associated with increased numbers of B cells, NKT cells, and inflammatory macrophages in the tumor microenvironment.

## Discussion

NOX1 and other NOX isoforms are involved in promoting tumor development (11, 12, 13, 14, 15, 16, 17). However, because NOX inhibitors tested in oncological models block several NOX isoforms, it was difficult to identify the NOX molecule most relevant for targeted tumor treatment. This is important because NOX family members can compensate for the decreased activity of a specific isoform, and this can cause downstream effects that counteract the desired therapeutic activity, including the induction of invasive phenotypes through the overexpression of metalloproteases (37). In addition, broad inhibition of multiple NOX isoforms may cause unwanted

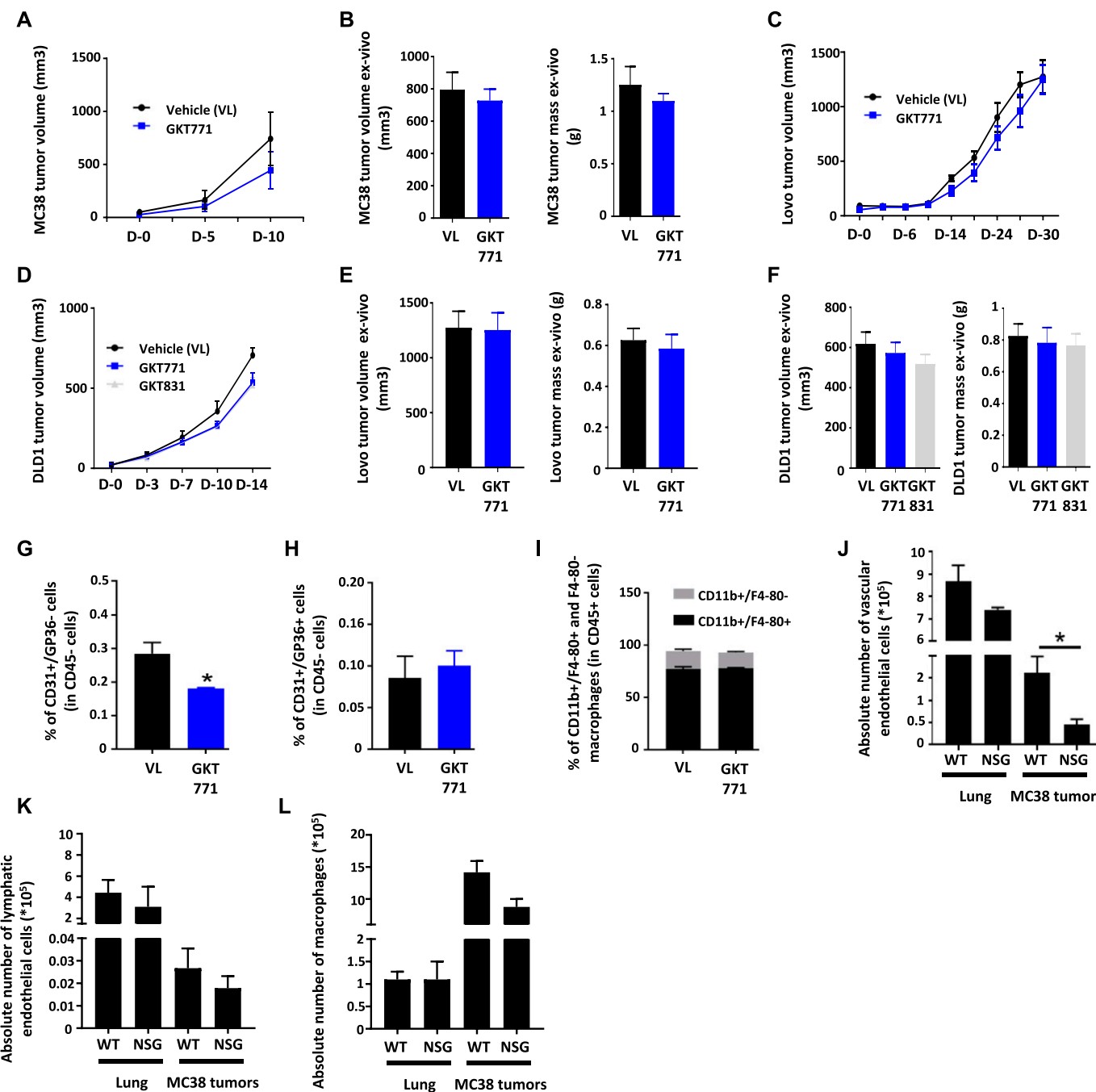

**Figure 4. The antitumor effect of GKT771 is suppressed in immunodeficient mice.**
**(A)** Immunodeficient NSG mice with established MC38 mouse colorectal tumors were treated with GKT771 or vehicle (n = 7). Tumor sizes were measured in vivo with a caliper during the course of the experiment. **(B)** MC38 tumor size and mass were measured ex vivo at the end of the experiment. **(C)** Immunodeficient NSG mice with established human LoVo colorectal tumors were treated with GKT771 or vehicle (n = 8 per group) (C). **(D)** NSG mice with established human DLD1 colorectal tumors were treated with GKT771 (n = 21), GKT831 (n = 14), or vehicle (n = 19) (D). Tumor sizes were measured in vivo by a caliper during the course of the experiment. **(E, F)** Tumor size and mass were measured ex vivo at the end of the experiment. **(G–I)** Tumors from NSG mice bearing DLD1 colorectal tumors were isolated and blood vascular endothelial cells (CD45⁻/CD31⁺/GP38⁻), lymphatic endothelial cells (CD45⁻/CD31⁺/GP38⁺), and macrophages (CD45⁺/CD11b⁺/F4/80⁺) were analyzed by flow cytometry (n = 19 for the vehicle group and n = 21 for the GKT771 group). The percentages of blood vascular cells, lymphatic endothelial cells, and macrophages in GKT771 or vehicle treated mice are shown. **(J–L)** Absolute numbers of blood endothelial cells, lymphatic endothelial cells, and macrophages in the lungs and tumors from C57/BL6 and NSG mice analyzed by flow cytometry (n = 5).

systemic effects by reducing ROS production in cells of the innate immune system, leading to increased susceptibility to infectious diseases. One example is defective NOX2 leading to chronic granulomatous disease (38). To prevent these damaging effects, we developed a novel NOX1 inhibitor called GKT771, which possesses an excellent combination of selectivity and potency. Pharmacological

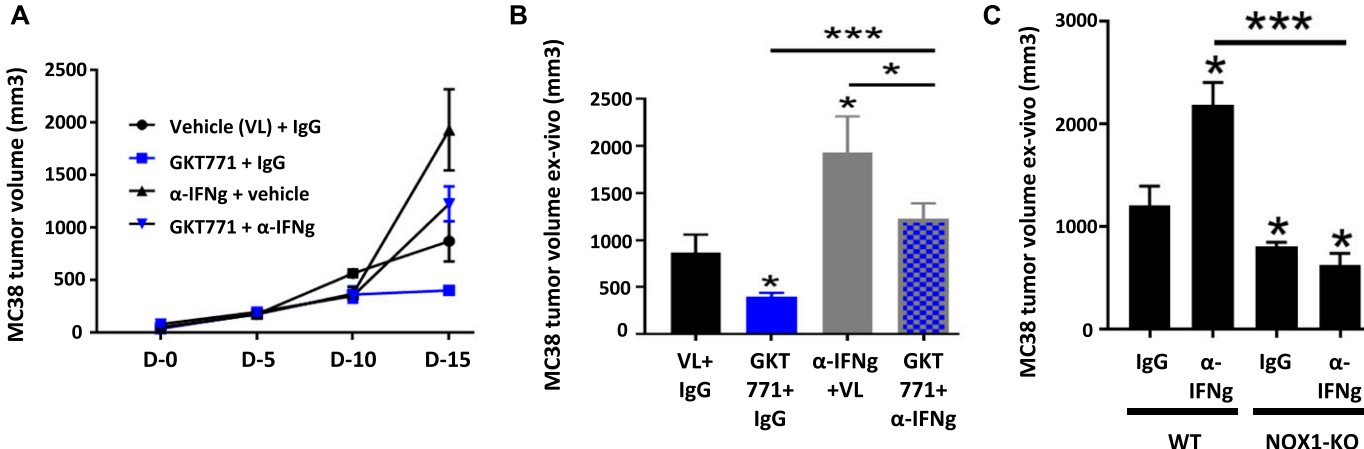

**Figure 5. The antitumor effect of NOX1 inhibition partially depends on IFN-γ.**
Mice with established MC38 tumors treated with GKT771, an anti–IFN-γ antibody, or combination thereof and the corresponding controls (n = 6/8 mice per group) are shown. **(A)** Tumor size was measured in vivo with a caliper during the course of the experiment. **(B)** Tumor size measured ex vivo at the end of the experiment. **(C)** C57/BL6 WT and NOX1-deficient mice with established MC38 tumors were treated with anti–IFN-γ antibody or the corresponding IgG (n = 8 mice per group). Tumor size was measured ex vivo at the end of the experiment.

in vitro assays using cell membranes and whole cell lysates confirmed that GKT771 blocks the activity of NOX1 without interfering with other NOX isoforms.

In this work, we investigated the efficacy of this novel compound in two experimental mouse tumor models. A colon carcinoma model was used because NOX1 is expressed at low level in normal colon epithelial cells and is highly up-regulated in colon adenocarcinoma, which, therefore, represents an ideal tumor target for colorectal cancer treatment (12). In addition, we used a melanoma model because melanoma cells tend to express low levels of NOX1 (data not shown) and may not respond to NOX1 inhibition. Collectively, we showed that NOX1 inhibition had potent antitumor effects in both models, and this effect was dependent on an intact immune system and required the inhibition of host NOX1.

The first important finding of the present work is that tumor NOX1 is not critical for the overall in vivo antitumor effects of GKT771. This is an unexpected result, as NOX1 gene silencing experiments demonstrated that NOX1 promotes colorectal carcinoma cell proliferation and cell cycle progression leading to increased tumor growth (15, 17, 39). In line with these results, in vitro treatment of cancer cells with GKT771 reduced tumor cell growth and cell cycle progression. However, the in vivo antitumor effect of GKT771 was not related to the presence or absence of NOX1 expression in cancer cells (i.e., DLD1 and LoVo cells). Rather, it was dependent on host NOX1 expression. Importantly, tissue expression of NOX1 in colon carcinoma patients has no prognostic value regarding survival (40). The observations that the therapeutic effects of GKT771 are dissociated from NOX1 expression levels in cancer cells seems at odd with previously published data. For this apparent discrepancy, we have no definitive explanation. However, there is a major difference between the experimental setting of these experiments: we acutely inhibited NOX1 function with a drug in established tumors, whereas the reported experiments were performed by injecting cancer cells with constitutively silenced NOX1 expression. The latter conditions may be more stringent than ours but does not reflect the clinical

conditions in which treated tumors are already well established. We will address this issue in future experiments, for example, with pharmacokinetics studies in tumor models. Indeed, it will be of interest to determine the amount of GKT771 present in tumors and healthy tissues/organs to assess its biodistribution and tumor penetration.

NOX1 inhibition in tumor cells in vitro decreased the production of VEGF-A and PLGF, two important angiogenic factors associated with tumor growth and progression (41, 42). We did not observe any inhibitory effect of GKT771 on VEGF-A mRNA expression levels, suggesting that the decreased VEGF-A secretion by cells in vitro due to NOX1 inhibition may involve post-translational regulation, such as protein maturation, transport, and secretion, or could be due to a decrease in cell number. However, although GKT771 reduced the plasma concentrations of PLGF and VEGF-A and suppressed blood vessel and lymphatic angiogenesis, this was not the main mechanism by which NOX1 suppressed tumor growth. NOX1-deficient mice showed reduced tumor angiogenesis, thereby validating the effects observed with pharmacological NOX1 inhibition but still supported tumor growth. GKT771 may inhibit angiogenesis through a dual mechanism: directly by suppressing endothelial proliferation and sprouting, and indirectly by affecting angiogenic factor production. The latter mechanism is supported by previous work showing that NOX1 activity contributes to VEGF-A expression and secretion. However, assessment of the effect of GKT771 in immunodeficient mice showed that despite significantly decreasing tumor angiogenesis (Fig 4J), GKT771 treatment was not effective at reducing tumor growth (Fig 4E–F). Thus, the anti-angiogenic effect of GKT771 is not sufficient to explain the global antitumor effects of NOX1 inhibition. This is the second important conclusion of our work. In parallel with the effects on angiogenesis, GKT771 decreased tumor lymphangiogenesis and reduced the concentration of the lymphatic growth factor VEGF-C in the plasma of tumor-bearing mice. A small, insignificant reduction of tumor lymphangiogenesis was observed in NOX1-deficient mice, suggesting the involvement of tumor-derived and NOX1-regulated factors. The lack of an

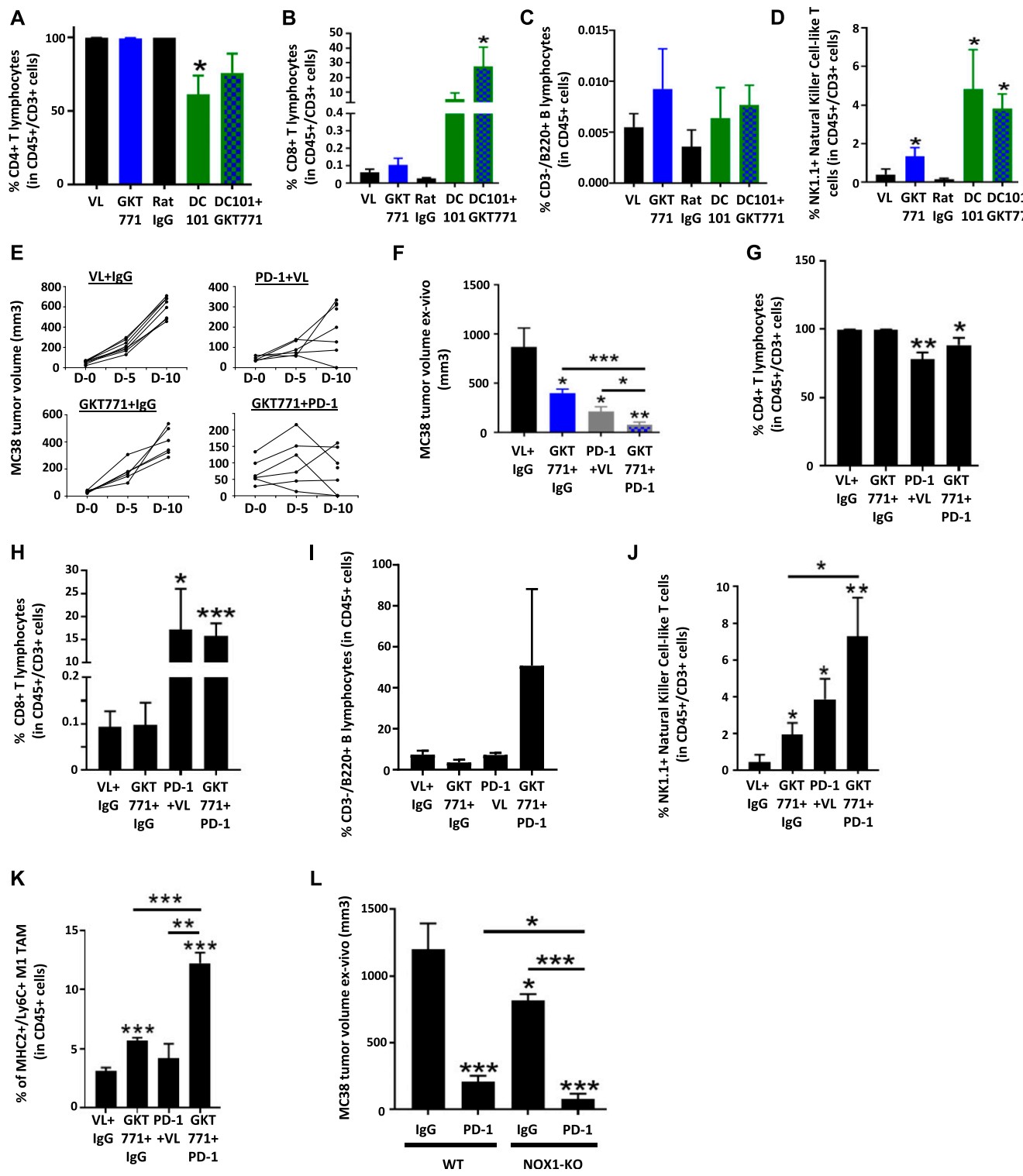

**Figure 6. GKT771 increases the efficacy of anti–PD-1 antibody immunotherapy in MC38 colorectal carcinoma.**
**(A–D)** MC38 tumors of C57/BL6 mice were isolated, stained with the indicated antibodies, and analyzed by flow cytometry. **(A–D)** CD4+ and CD8 T+ lymphocytes
([CD4+/CD3+/CD45+] or [CD8+/CD3+/CD45+]), (C) B lymphocytes (CD45+/CD3−/B220+), and (D) NKT cells (CD45+/CD3+/NK1.1+). The percentages of cells in the differently
treated groups are shown. **(E, F)** Mice with established MC38 tumors were treated with GKT771 (n = 6), anti-PD1 antibody (n = 7), combinations thereof (n = 6), or the
corresponding controls (n = 8) are shown. **(E, F)** Tumor size was measured in vivo with a caliper during the experiment. **(G)** Tumor size was measured ex vivo at the end of
the experiment. **(G–K)** Tumors were isolated, and the percentages of CD4+ T lymphocytes (G), CD8+ T lymphocytes (H), B lymphocytes (I), NKT cells (NKT) (J),
and proinflammatory macrophages (K) were determined by flow cytometry. C57/BL6 WT and NOX1-deficient mice with established MC38 tumors were treated with anti-PD1
antibody or the corresponding IgG (n = 6/8 mice per group). **(L)** Tumor size was measured ex vivo at the end of the experiment (L).

additional GKT771 effect on angiogenesis in NOX1-deficient mice confirmed the specificity of GKT771 for NOX1 and supported that NOX1 expressed by host cells constitutes a major target of GKT771. As tumor angiogenesis and lymphangiogenesis are important local events that promote regional and systemic dissemination (43, 44), in future studies, it will be important to test whether these effects of GKT771 translate into reduced regional and distant metastases.

The third important finding of this study is that the antitumor effect of NOX1 requires an intact immune system. Host immune cells in tumors contribute to antitumor effects in many types of tumors (45). In contrast to the antitumor effect observed in the immunocompetent syngeneic murine model, GKT771 had no antitumor effects in immunodeficient mice bearing human or mouse CRCs. This suggests that the immune system provides an essential contribution to the antitumor effects of NOX1 inhibitors. NOX1 is expressed by various immune cell populations, including macrophages and lymphocytes (46). In addition, numerous inflammatory cytokines such as INF-$\gamma$ affect the expression levels of NOX1, and INF-$\gamma$ itself is involved in the recruitment of hemopoietic cells (47), suggesting that NOX1 contributes to immune cell recruitment in the tumor microenvironment. Macrophages are among the immune cells whose phenotype is regulated by NOX1 expression. Double deletion of NOX1 and NOX2 induces polarization of macrophages towards the M1 proinflammatory phenotype (22). However, whether deletion of NOX1 alone is sufficient to induce M1 polarization remains unclear. To investigate this hypothesis, we used different cellular markers to quantify the number of total macrophages (CD11b$^+$/F4-80$^+$), TAMs (CD11b$^+$/CD68$^+$/F4-80$^+$), or proinflammatory, M1 like macrophages (MHC II$^+$/Ly6C$^+$) in tumors treated with GKT771 or tumors growing in NOX1-deficient mice. Interestingly, our results showed that blocking NOX1 increased the number of proinflammatory macrophages in the tumors. This effect was also observed by genetic deletion of NOX1 in the host. In addition to its effect on macrophages, GKT771 increased the number of NKT cells infiltrating the tumor. This finding is interesting as NKT cells have antitumor activity and can activate immune cells, particularly, CD8$^+$/CD4$^+$ T and B lymphocytes, TAMs, and also NK cells (32). Of further interest is the fact that combined GKT771 and DC101 treatment increased CD8$^+$ T cell recruitment to the tumor. The critical role of the immune system in the GKT771 antitumor activity is further supported by the observation that blocking IFN-$\gamma$ with a monoclonal antibody partially reduced the efficacy of GKT771 therapy. This is consistent with the previously reported correlation between NOX1 and IFN-$\gamma$ expression in colon epithelial cells (28).

The fourth important result was that blocking the T-cell checkpoint inhibitor PD-1 with an anti-PD1 antibody in combination with GKT771 enhanced the antitumor activity of these compounds as single agents. Such combinatorial use of the anti-PD1 antibody with other antitumor drugs targeting different molecular mechanisms was reported for other therapeutic compounds. For example, anti-PD1 treatment combined with anti–angiopoietin-2 or anti-CSF1 antibodies suppresses tumor development in several mouse models (48, 49). It will be of clinical relevance to test whether tumors refractory to anti-PD1 therapy may be sensitized by NOX1 inhibition.

In conclusion, we showed that blocking NOX1 with the novel small-molecule inhibitor GKT771 inhibits tumor growth in mice. We showed that host NOX1 is critical for this effect. Stimulation of the innate immune system, in particular, proinflammatory macrophages and NKT cells, and inhibition of angiogenesis and lymphangiogenesis are events associated with the antitumor activity of GKT771. As the antitumor effects of GKT771 requires an intact immune system, we propose that GKT771 elicit immunomodulatory effects essential for its antitumor activity. Consistent with this notion, combination of GKT771 with the immune cell checkpoint inhibitor anti-PD1 resulted in a mutual enhancement of the antitumor activity of each single agent. NOX1 inhibition, alone or in combination with checkpoint inhibitors, may, therefore, represent a promising antitumor strategy in colorectal cancer (Fig 7).

# Materials and Methods

### Inhibitors

GKT771 and GKT831 NOX inhibitors were developed by GenKyoTex S.A. using recombinant cells transfected with the NOX1 to NOX5 isoforms. Activity of compounds against the different NOX isoforms was measured as previously described (50). Briefly, membranes prepared from the different recombinant cell lines were incubated in PBS with Amplex Red, HRP, and appropriate cofactors. ROS production was induced by the addition of NADPH. Inhibitory activity of compounds was measured in the presence of increasing concentrations ranging from 1 nM to 100 $\mu$M. After 20 min of incubation at 37°C, ROS levels were measured using a BMG Labtech microplate reader.

### Cell culture

Mouse primary lung microvascular endothelial cells (mMVEC-L) were grown in complete mouse endothelial cell medium using a kit (M1168-Kit; Cell Biologics). Human dermal lymphatic endothelial cells were grown in Endothelial Cell Growth Medium from PromoCell. Human colon cancer cell lines SW620 (CVCL_0547), DLD1 (CVCL_0248), Caco-2 (CVCL_0025), and LoVo (CVCL_0399) were cultured in DMEM (Life Technologies) supplemented with FCS (10%), PS (1%), L-glutamine (1%), and sodium pyruvate (1%). Mouse immortalized lymphatic endothelial cells (LyEnd.1), mouse melanoma cancer cell lines B16-F10 (CVCL_0159), and B16-F0 (CVCL_0604) and the mouse colorectal cancer cell line MC38 (CVCL_B288) were cultured in DMEM (Life Technologies) supplemented with FCS (10%), PS (1%), L-glutamine (1%), and sodium pyruvate (1%). The cells were grown in a humidified atmosphere with 5% $CO_2$ at 37°C. Absence of mycoplasma contamination from the different cell lines used during the experiments was performed by using the PCR mycoplasma Test Kit I/C (PromoCell).

### BrdU incorporation assay

mMVEC-L cells were treated with DMSO as vehicle or GKT771 for 1 h at 37°C. Cells were stimulated or not with recombinant murine VEGF (Catalog No. 450-32 – Lot #010899; Peprotech) at 20 ng/ml for 24 h at 37°C.

**Life Science Alliance**

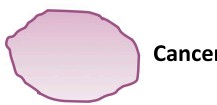

**Legend:**

 Cancer cells

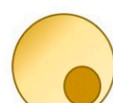 Blood vessels

Lymphatic vessels

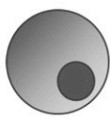 CD4⁺ T lymphocyte

CD8⁺ T lymphocyte

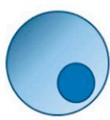 B lymphocyte

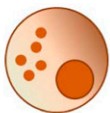 NKT cells

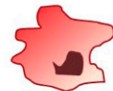 Pro-inflammatory macrophages

| Non-treated tumors | NOX1 targeting |
|---|---|
| VEGF-R2 targeting | NOX1 and VEGF-R2 targeting |
| PD-1 targeting | NOX1 and PD-1 targeting |

The cells were then labeled with BrdU for 2 h at 37°C, after which the medium was removed and the cells were fixed by addition of FixDenat solution for 30 min at room temperature. After FixDenat removal, the cells were incubated with anti–BrdU-POD for 1.5 h at room temperature. The cells were then washed three times for 5 min at room temperature, incubated with the substrate for 3 min at room temperature on a shaker, and chemiluminescence was measured using a fluorimeter within 10 min after adding the reagent.

## Cell proliferation assay

Cells ($10^6$) were plated in six-well plates overnight followed by drug exposure for different time periods. Adherent cells were detached from the plate using trypsin, followed by addition of PBS without $Ca^{2+}/Mg^{2+}$. Suspended cells were centrifuged at 1,500 rpm for 5 min by using AWEL centrifuge (C48) and swing out rotor (SO400), and resuspended in PBS/10% FCS before counting. Dead and living cells were counted with the Countess II FL Automated Cell Counter (Thermo Fisher Scientific) after adding trypan blue.

## ELISA

VEGF-A, VEGF-C, and PLGF measurement in platelet-diminished plasma was performed using mouse blood subjected to two steps of centrifugation (3,500 rpm for 15 min) and cell culture supernatant prepared using one centrifugation step (10,000 rpm for 10 min). ELISA was performed using the mouse VEGF-A/mouse PLGF DuoSet ELISA (R&D Systems) or the mouse VEGF-C ELISA kit from CusaBio following the manufacturer's protocols.

## Quantitative polymerase chain reaction

Cell line and tumor RNA was extracted using the RNeasy Mini kit (QIAGEN) according to the supplier's instructions. The extracted RNA was assayed using NanoDrop at 260 nm (Thermo Fisher Scientific). RT was performed by adding 1 $\mu$g of RNA to 10 $\mu$l of a mixture (RT buffer DNX 25×, 100 mM; RT random primers 10×; MultiScribe Reverse Transcriptase; and $H_2O$ [Applied Biosystems]). RT was performed in a thermocycler at 25°C for 10 min, 37°C for 2 h, 85°C for 5 min, and then 4°C. cDNA (50 ng/$\mu$l) was mixed with 24 $\mu$l of a mixture of SYBR Green (Agilent, Stratagene), right and left primers, and $H_2O$. PCR was then performed using the Stratagene Mx 3000 Pro QPCR (Agilent Technologies) according to the following profile: one cycle of 10 min at 95°C, and then 40 cycles of 30 s at 95°C and 1 min at 60°C. The results were analyzed using MxPro software (Stratagene). Human GAPDH (NM_002046.3; AB Applied Biosystems) was used as an internal control. The qRT-PCR were routinely performed with two technical replicates and three biological replicates. The qRT-PCR data were presented as delta Ct values of the investigated genes relative to GADPH.

## Animal procedures

C57Bl6-J immunocompetent and NSG immunodeficient (NOD.Cg-$Prkdc^{scid}$ $Il2rg^{tm1Wjl}$/SzJ) mice were purchased from The Jackson Laboratory and bred in-house. NOX1-deficient mice (B6.129X1-Nox1tm1Kkr/J) were purchased from The Jackson Laboratory. All animal procedures were performed in accordance with the Institutional Ethical Committee of Animal Care in Geneva and the Swiss Cantonal Veterinary Office (Authorization numbers: Ge93/14 and Ge17/16). Male and female mice between 5 and 7 wk were used.

## Mouse aortic ring assay

The thoracic aorta was dissected from 6–8-wk-old C57BL/6 mice, NSG mice, and NOX1 KO mice (51). Adipose tissue surrounding the aorta was removed. After the cleaning step, the aortas were cut into 0.5-mm wide rings with a scalpel, and the rings were then incubated in serum-free OptiMEM (Life Technologies) overnight at 37°C in 5% $CO_2$. The next day, the rings were embedded in 1 mg/ml type I collagen (Millipore) and maintained at 37°C in a 5% $CO_2$, 95% humidity incubator in OptiMEM supplemented with 2.5% FCS, 100 U/ml penicillin/streptomycin, and 30 ng/ml VEGF165 (Peprotech). Aortic rings were cultured in the presence of the different NOX inhibitors (GKT771 and GKT831). Microvessel outgrowth was imaged daily by phase-contrast microscopy (ImageXpress; Molecular Devices) and subsequently quantified with MetaMorph (Molecular Devices). The nonsprouting fragments were excluded from the quantification.

## Tumor growth studies

Mouse and human tumors were produced by subcutaneous injection of tumor cells diluted in PBS ($1 × 10^6$ for DLD1, $5 × 10^5$ for MC38, and $1 × 10^6$ for B16-F10 and LoVo). When tumors reached 50 or 200 mm$^3$ depending of the experiments, the mice were subjected to intraperitoneal administration of purified antibodies twice per week at a dose of 600 $\mu$g per injection per mouse for DC101; gavage twice daily for the NOX1 inhibitor GKT771 at 10 mg/kg, or once daily for the NOX1/4 inhibitor GKT831 at 20 mg/kg; or Vehicle (VL) (i.e., methylcellulose and Tween 80) until the mice were euthanized. Tumor size was measured with a caliper and tumor volume was determined according to the following equation: (length × width × thickness) × 0.5236. Tumor size and mass were also evaluated ex vivo at the end of the experiment after the animals were euthanized. Before being euthanized, the animals were anesthetized with a mix of ketamine and xylasizne (80 mg/kg and 10 mg/kg, respectively) before intratracheal instillation of 10% formalin to fix tumors and organs. After paraffin embedding, the samples were cut into sections at five distinct levels. For flow cytometry analysis, tumors were removed without fixation with PFA.

---

**Figure 7.   Schematic model proposed for the impact of NOX1 and NOX1-combined treatments on tumor host cell composition.**
The composition in tumor-associated host blood vessels, lymphatic vessels, and immune cells of nontreated tumors and corresponding treated tumors is represented. The amount of tumor-associated blood and lymphatic vessels is decreased in treated tumors compared with nontreated tumors. The infiltration of proinflammatory and antitumor immune cells is increased in NOX1, VEGFR-2, and PD-1 monotherapy treatedtumors. These effects are more pronounced in NOX1/VEGFR-2 and NOX1/PD-1 combined therapy compared with each treatment alone and nontreated tumors.

All therapeutic monoclonal antibodies (mAbs), including anti-mouse PD-1 (clone RMPI-14 IgG2a; 10 mg/kg) and anti–IFN-γ (clone XMG1.2 IgG1; 10 mg/kg), were obtained from BioXCell and administered two times per week.

Macrophage depletion in tumors was achieved by injection of mice with clodronate liposomes two times per week. Clodronate liposomes and control liposomes were obtained from LIPOSOMA.

## Flow cytometry

After dissection, the tumors and lungs were cut with razor blades and digested using gentleMACS Dissociators with the corresponding mouse and human dissociation kit from Miltenyi Biotec following the manufacturer's protocols. Single-cell suspensions for both tumor and lung samples were obtained by straining through a 70-$\mu$m mesh filter, and the cells were washed twice in FACS buffer (PBS/Fetal Calf Serum 5%/5 mM EDTA). The cells were incubated with fluorescent antibodies for 30 min at 4°C to prevent Fc receptor binding and washed once with FACS buffer. Then, the cells were stained with the indicated fluorophore-conjugated antibodies and analyzed using the Gallios flow cytometer (Beckman Coulter) or sorted with the FACSARIA II flow cytometer (BD).

## In vivo angiogenesis assay

Angiogenesis was assessed in male C57BL/6 mice (20–22 g) ordered from Elevage Janvier. Angioreactors ordered from Amsbio (directed in vivo angiogenesis assay ref 3450-048-K) were prepared according to the kit instructions. Briefly, implant-grade silicone cylinders closed at one end, called angioreactors, are filled with 20 $\mu$l of Trevigen's PathClear basement membrane extract premixed with or without angiogenic-modulating factors. A mix of VEGF (10 $\mu$g) and FGF (50 $\mu$g) obtained from PeproTech was used. Two angioreactors per mouse were then implanted subcutaneously in the dorsal flank of mice. At the onset of angiogenesis, vascular endothelial cells grow into the basement membrane extract and form vessels in the angioreactor. As early as 15 d postimplantation, there were enough cells to determine an effective dose response to angiogenic-modulating factors using an FITC-lectin detection system. Mice were treated with GKT771 by oral gavage (10 ml/kg/d) from D-0 to D-14.

## Statistical analysis and expression of results

Data were expressed as mean values ± SEM of the indicated number of experiments. Statistical analysis was performed with Prism software (GraphPad Software Inc.). The variance between different groups was estimated before statistical analysis. When comparing more than two groups, nonparametric one-way ANOVA followed by Dunn's multiple comparison test was used. Significant differences between two groups were determined using the unpaired $t$ test or Mann–Whitney test. A $P$-value ≤0.05 was considered significant. In animal studies, the investigator was blinded to the group allocation and mice were distributed at random into separate groups.

# Supplementary Information

# Acknowledgements

This work was supported by grants from the Swiss National Science Foundation (310030_153456) and ONCOSUISSE and Sigrid Juselius Foundation to BA Imhof, the Medic Foundation and the Swiss National Science Foundation (31003A_135738 and 31003A_179248) to C Ruegg.

## Author Contributions

J Stalin: conceptualization, data curation, formal analysis, validation, investigation, visualization, methodology, project administration, and writing—original draft, review, and editing.
S Garrido-Urbani: data curation, formal analysis, investigation, and methodology.
F Heitz: data curation, formal analysis, investigation, and methodology.
C Szyndralewiez: data curation, formal analysis, investigation, and methodology.
S Jemelin: data curation, formal analysis, and investigation.
O Coquoz: data curation, formal analysis, investigation, and methodology.
C Ruegg: conceptualization, supervision, funding acquisition, validation, investigation, visualization, methodology, project administration, and writing—original draft, review, and editing.
BA Imhof: conceptualization, supervision, funding acquisition, validation, investigation, visualization, methodology, project administration, and writing—original draft, review, and editing.

## Conflict of Interest Statement

F Heitz and C Szyndralewiez are employed by GenKyoTex. The other authors declare no conflicts of interest.

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
