## [Reviewer comments · Life Science Alliance]

Life Science Alliance

Inhibition of host NOX1 blocks tumor growth and enhances checkpoint inhibitor-based immunotherapy

Jimmy Stalin, Sarah Garrido-Urbani, Freddy Heitz, Cédric Szyndralewicz, Stephane Jemelin, Oriana Coquoz, Curzio Ruegg, and Beat Imhof

DOI: <https://doi.org/10.26508/lsa.201800265>

Corresponding author(s): Jimmy Stalin, Faculty of Science and Medicine University of Fribourg; Beat Imhof, Geneva, University of; and Curzio Ruegg, University of Fribourg

Review Timeline:

Submission Date:	2018-12-06
Editorial Decision:	2019-01-03
Revision Received:	2019-04-27
Editorial Decision:	2019-05-17
Revision Received:	2019-06-13
Accepted:	2019-06-13

Scientific Editor: Andrea Leibfried

Transaction Report:

January 3, 2019

Re: Life Science Alliance manuscript #LSA-2018-00265-T

Dr. Jimmy Stalin
Faculty of Science and Medicine University of Fribourg
Department of Oncology/Microbiology/Immunology (OMI)
Ch. du Musée 18, PER17
Fribourg, Canton de Fribourg 1700
SWITZERLAND

Dear Dr. Stalin,

Thank you for submitting your manuscript entitled "Inhibition of host NOX1 blocks tumor growth and enhances checkpoint inhibitor-based immunotherapy" to Life Science Alliance. The manuscript was assessed by expert reviewers, whose comments are appended to this letter.

As you will see, the reviewers think that your work is of interest to others. However, they also note several issues that need to get addressed prior to publication. We would thus like to invite you to provide a revised version of this work. Reviewer #1's concerns seem straightforward to address. Reviewer #2 raises multiple concerns. While the requested in-depth mechanistic insight does not need to get addressed for publication here, we expect that the revision addresses this reviewer's request for better controlling/analyzing Nox expression levels and to represent the data differently / toning-down the conclusions on tumor microenvironment versus tumor intrinsic effects as well as on the effects on tumor progression.

Thank you for this interesting contribution to Life Science Alliance. We are looking forward to

receiving your revised manuscript.

Sincerely,

- A letter addressing the reviewers' comments point by point.
- An editable version of the final text (.DOC or .DOCX) is needed for copyediting (no PDFs).
- High-resolution figure, supplementary figure and video files uploaded as individual files: See our detailed guidelines for preparing your production-ready images, <http://life-science-alliance.org/authorguide>
- Summary blurb (enter in submission system): A short text summarizing in a single sentence the study (max. 200 characters including spaces). This text is used in conjunction with the titles of papers, hence should be informative and complementary to the title and running title. It should describe the context and significance of the findings for a general readership; it should be written in the present tense and refer to the work in the third person. Author names should not be mentioned.

B. MANUSCRIPT ORGANIZATION AND FORMATTING:

Full guidelines are available on our Instructions for Authors page, <http://life-science-alliance.org/authorguide>

Reviewer #1 (Comments to the Authors (Required)):

This manuscript provides diverse and high quality data demonstrating that pharmacologically targeting NOX1 in mouse models can restrict the development of colorectal cancer. This was further corroborated by use of a genetic mouse model lacking NOX1. The data are generally compelling and clearly presented, and fully support the central claims of the manuscript. The use of multiple mouse models is a particular strength. Two important points for clarification are outlined below.

1. The Investigators should provide evidence that GKT771 is a highly selective inhibitor of NOX1 and does not inhibit other NOX isoforms, if this has not been published previously in the academic setting.

2. The Investigators use CD45/CD31/Gp38 combinations to specifically detect blood vascular versus lymphatic endothelial cells by flow cytometry. Where is the evidence that these combinations are appropriate for this task? This is an important issue for the findings about tumor angiogenesis and lymphangiogenesis in the mouse models.

Reviewer #2 (Comments to the Authors (Required)):

This manuscript by Stalin and colleagues provides interesting information regarding the role of NOX1 in the tumor microenvironment on the growth of murine and human tumor cell lines propagated as xenografts. The most important contribution is the demonstration that proliferation of murine colon and murine melanoma xenografts are diminished in NOX1 knockout mice; and that this observation is mimicked, in part, by administration of the putative NOX1 inhibitor GKT771. Experiments demonstrating that a PD-1 immune checkpoint inhibitor enhances the effects of GKT771 in murine tumor cell xenografts carried by immunocompetent mice, in the face of increased tumor associated macrophages in these xenografts, is also of real interest. However, some of the conclusions drawn by the authors exceed the data provided; these conclusions need to be tempered by additional experiments that might (or might not) support their hypotheses. The issues that need to be addressed include the following:

1. An important assertion in the manuscript is that the intrinsic NOX1 expression of the tumors studied was not the target of GKT771. The effect of NOX1 knockout/chemical inhibition is presumed to be related to one or more features of the tumor microenvironment.

a. What are the expression levels of the 7 NOX isoforms in MC38 colon cancer cells and in the B16 melanoma line both in vitro and when grown as xenografts? Cell lines devoid of NOX expression have been shown to upregulate various NOX species during passage as xenografts in vivo. It cannot be assumed that just because MC38 is a colon cancer, it must have elevated NOX levels; nor can the converse be assumed for B16; nor can the expression levels in vitro be assumed to be unchanged in vivo. Furthermore, the authors have utilized a series of human colon cancer cell lines to support the contention that NOX inhibition by GKT771 is ineffective in an immunocompromised host. However, the Lovo cell line lacks functional NOX1 in vitro as does the DLD1 line; hence, the data shown in Figs. 4C and D do not address the issue of the role of NOX1 intrinsic to the tumor as a target for GKT771 (Page 9). On the other hand, genetic knockdown of NOX1 in Caco-2 and HT-29 human colon cancer cells in vitro-cells which do possess functional NOX1 activity-clearly demonstrates inhibition of tumor cell growth (and apoptosis for Caco-2 cells) produced by a major block in cell cycle progression at G1 (Int. J. Cancer 128: 2581-2590, 2011). Recent experiments have also shown that NOX1 knockdown for cells that possess functional, ROS-producing NOX1 markedly decreases tumor cell growth both in vitro and in vivo (in nu/nu mice).

b. Absent functional NOX1 expression, the in vitro results presented in Supplemental Fig. 3E, are difficult to interpret. What is the mechanism by which GKT771 inhibits tumor cell or, for that matter, endothelial cell proliferation (Fig. 1J) in vitro? While some potential mechanisms of in vivo growth inhibition are addressed in the paper, there is essentially no such data provided for the in vitro studies-which has major impact for understanding the results presented.

2. With respect to experimental methodology, the authors have chosen to treat tumors in vivo at a size (50 mm³) that is essentially unmeasurable. Put another way, the results demonstrate an adjuvant effect rather than a therapeutic effect on well-established tumors. This is a very important distinction in terms of how the data are to be interpreted; the paper would be much stronger if the effects of GKT771 on tumors that can be measured (≥ 200 mm³) were assessed, since this would have major implications regarding how one might think about the value of NOX1 inhibition in various clinical situations, and the potential therapeutic value of GKT771 in particular.

3. What are the pharmacokinetics of GKT771 in vivo in these tumor-bearing models? Do the in vivo drug levels come anywhere near the micromolar levels needed for growth inhibition and anti-angiogenic effects in vitro?

4. Do the authors have data to address the mechanism by which GKT771 diminishes VEGF secretion in vitro? Is it simply a result of decreased tumor cell proliferation?

5. One explanation for the synergistic effects of GKT771 and a PD-1 blocker is that the small molecule might enhance the mutational burden (and hence antigen presentation) of the tumor cells (as has been shown for treatment with temozolamide). Do the authors have any information on potential non-NOX1 related effects of GKT771 that might be consistent with this current hypothesis that underlies much of what is known regarding the mechanism of action of PD-1 antibodies?

6. As noted above, it is difficult to conclude that the intrinsic NOX1 levels of the tumor are unrelated to the apparent effects on the microenvironment when essentially all of the tumor cell lines studied either lack functional NOX1 or are likely to do so (Page 13); similarly, since tumor growth delay has been observed secondary to NOX1 knockdown in immunocompromised mice previously, it is quite difficult to conclude that antitumor effects of NOX1 inhibition require an intact immune system. Rather, it would appear that an intact innate immune system may play a role in NOX1-related tumor cell growth. In particular, the authors should be aware that the partial growth delay observed for GKT771 treatment in both the B16 and MC38 systems was modest (tumors continued to grow over the 15 days of observation). Since larger tumors were not used, it is unclear whether the agent would cause tumor regressions per se.

7. The Methods state that experiments were also performed with SW480 and HCT-116 cell lines; what were those results?

Reviewer 1:**1. The Investigators should provide evidence that GKT771 is a highly selective inhibitor of NOX1 and does not inhibit other NOX isoforms, if this has not been published previously in the academic setting.**

Indeed, evidence that GKT771 is a selective NOX1 inhibitor was not explicitly included in the manuscript. The data of the characterization of GKT771 have been deposited in a patent in a publicly available patent data base (WO 2016/098005). Our partner, Genkyotex S.A., is currently in the process of publishing these data in a pharmacological/chemical journal. As this paper is not published yet, we agree to provide the necessary details about the selectivity of GKT771 to you and the referees as would be useful and important information.

To characterize the specificity of NOX inhibitors in vitro, Genkyotex S.A. used CHO cells transfected with the different NOXs isoform (NOX1 to 5 and Xanthine oxidase). As a readout of activity, they measured ROS, H₂O₂ and O₂⁻ production using chemical probes in purified cell membrane lysates in the presence or absence of treatment with GKT771 (or other inhibitors).

Using this assay, they could demonstrate that GKT771 specifically inhibits production of ROS and H₂O₂ in cell membrane lysates of NOX1 overexpressing CHO cells compared to membrane lysates expressing other NOXs isoforms. Moreover, O₂⁻ production in this model is also specifically decreased in NOX1 overexpressing CHO cell membrane lysate compared to NOX2-overexpressing CHO cell membrane. The K_i and E_{max} for each assay are summarized in the attached table below (Fig. 1). These results demonstrate that GKT771 effectively and specially inhibit NOX1.

Figure 1: ROS, H₂O₂ and O₂⁻ detection in membrane lysate of CHO cells expressing NOXs isoform and summary table of Ki and Emax of GKT771.

As Genkyotex is in the process of publishing these results in a separate original paper, we cannot include them in the present manuscript. Nevertheless, in the materials and methods section, we have added the following paragraph:

Inhibitors

GKT771 and GKT831 NOX inhibitors were developed at Genkyotex S.A using CHO cells transfected with the NOX1 to NOX5 isoforms and Xanthine oxidase. As a readout of activity, ROS, H₂O₂ and O₂⁻ production were measured using chemical probes in purified cell membrane lysates in the presence or absence of treatment with inhibitors. The characterization of GKT771 is publicly available under patent WO 2016/098005.

2. The Investigators use CD45/CD31/Gp38 combinations to specifically detect blood vascular versus lymphatic endothelial cells by flow cytometry. Where is the evidence that these combinations are appropriate for this task? This is an important issue for the findings about tumor angiogenesis and lymphangiogenesis in the mouse models.

We agree with this reviewer that the distinction between blood vascular versus lymphatic endothelial cells is of primary interest and major importance in these experimental models. In most reports tumor angiogenesis is “quantified” by CD31 (PECAM) immunohistochemistry staining of tumor sections. Alternatively, CD31 expressing cells can be quantified in a tumor homogenate by flow cytometry analysis. CD31 is expressed by both lymphatic and vascular endothelial cells, while podoplanin (gp38) is expressed in lymphatic endothelial cells but not in blood vascular endothelial cells [1-2]. Thus, CD31/GP38 double staining is used to distinguish between tumor blood vascular (CD31+/Gp38-) and lymphatic (CD31+/Gp38+) endothelial cells by flow cytometry experiments. CD45 staining was performed to exclude hematopoietic cells, monocytes in particular, that also express CD31. Thus, we considered

CD45 negative (non-hematopoietic cells), CD31 positive and gp38 negative cells as blood endothelial cells and CD45 negative, CD31 and gp38 double positive cells as lymphatic endothelial cells. This flow cytometry staining strategy to distinguish these two cell types has been previously reported in the literature [3].

References:

1/ *Podoplanin a specific marker for lymphatic endothelium expressed in angiosarcoma. Breiteneder-Geleff S, Soleiman A, Horvat R, Amann G, Kowalski H, Kerjaschki D. Verh Dtsch Ges Pathol. 1999; 83: 270-5.*

2/ *Plasticity of blood- and lymphatic endothelial cells and marker identification. Keuschnigg J, Karinen S, Auvinen K, Irjala H, Mpindi JP, Kallioniemi O, Hautaniemi S, Jalkanen S, Salmi M. PLoS One. 2013 Sep 10; 8(9):e74293.*

3/ *Consensus guidelines for the use and interpretation of angiogenesis assays. Nowak-Sliwinska P et al., Angiogenesis. 2018 Aug; 21(3):425-532].*

Reviewer 2:

1. An important assertion in the manuscript is that the intrinsic NOX1 expression of the tumors studied was not the target of GKT771. The effect of NOX1 knockout/chemical inhibition is presumed to be related to one or more features of the tumor microenvironment.

a. What are the expression levels of the 7 NOX isoforms in MC38 colon cancer cells and in the B16 melanoma line both in vitro and when grown as xenografts? Cell lines devoid of NOX expression have been shown to upregulate various NOX species during passage as xenografts in vivo. It cannot be assumed that just because MC38 is a colon cancer, it must have elevated NOX levels; nor can the converse be assumed for B16; nor can the expression levels in vitro be assumed to be unchanged in vivo.

In the manuscript, we demonstrate that NOX1 expressed by the tumor cells is not the main target for the anti-tumor effects of GKT771 treatment while NOX1 expressed in the host and the immunological mouse background have a dominant impact. GKT771 may as well inhibit NOX1 expressed in cancer cells but this inhibition, in absence of immune cells or stromal cells expressing NOX1, is not sufficient to reduce tumor growth. Host NOX1 expression and its inhibition are determinant for the response.

Previously, expression of NOXs isoform and especially NOX1 (i.e. the GKT771 target), have been studied and reported in human cancer cell lines but not in the mouse cell lines used in our study. We therefore agree that the reviewer raised a relevant point.

To address this reviewer's questions, we performed qPCR experiments to detect basal levels of expression of six NOXs isoforms in B16F10 and MC38 cancer cell lines in vitro and in lysates of MC38 and B16F10 tumors grown in mice. The NOX5 isoform present in human, but absent in rodents could not be analyzed in these experiments.

Basal expression of mRNA for the NOX1, NOX4 and DUOX2 isoforms was detected in both cell lines in vitro. Interestingly, MC38 expresses NOX1 mRNA at higher levels compared to B16F10 while the opposite is true for NOX4 expression. DUOX2 expression was similar for both cell lines (Fig. 2). Moreover,

we also found NOX3 and DUOX1 expression in B16F10 cells whereas no expression of NOX2 was detected in both cell lines.

Figure 2: NOXs mRNA expression in MC38 and B16F10 (Ct NOXs-Ct GAPDH).

The mRNA expression analysis of tumors demonstrates that NOX1, DUOX1 and DUOX2 displayed similar expression patterns in vivo and in vitro. Also, MC38 and B16F10 tumors expressed similar levels of NOX4 mRNA.

Interestingly, we observed NOX2 and NOX3 expressing in both tumor types, while in vitro none of the cell lines did express NOX2, while NOX3 was only expressed by B16F10 cells. This difference could be due to the presence of stromal cells expressing these NOXs (the lysates contain both tumor and stromal cells) or to an upregulation in cancer cells when they are growing as a tumor (Fig.3).

Figure 3: NOXs mRNA expression in MC38 and B16F10 tumors (Ct NOXs-Ct GAPDH).

These results are now included in the revised manuscript in the Supplementary Fig. 1A and 1B and in the text:

In these two models NOX1 and NOX4 expression was different among the cell lines and conditions: NOX1 expression was higher in B16F10 cells both in vitro and in vivo, while NOX4 expression was higher in MC38 cells in vitro but similar to B16F10 in vivo (Supplementary Fig. 1A and 1B).

Furthermore, the authors have utilized a series of human colon cancer cell lines to support the contention that NOX inhibition by GKT771 is ineffective in an immunocompromised host. However, the Lovo cell line lacks functional NOX1 in vitro as does the DLD1 line; hence, the data shown in Figs. 4C and D do not address the issue of the role of NOX1 intrinsic to the tumor as a target for GKT771

(Page 9). On the other hand, genetic knockdown of NOX1 in Caco-2 and HT-29 human colon cancer cells in vitro-cells which do possess functional NOX1 activity-clearly demonstrates inhibition of tumor cell growth (and apoptosis for Caco-2 cells) produced by a major block in cell cycle progression at G1 (Int. J. Cancer 128: 2581-2590, 2011). Recent experiments have also shown that NOX1 knockdown for cells that possess functional, ROS-producing NOX1 markedly decreases tumor cell growth both in vitro and in vivo (in nu/nu mice).

We used human and mouse MC38 cancer cell lines in immunocompromised mice to address the hypothesis that host NOX1-expressing cells may constitute a target of GKT771. Based on the following observations we concluded that an immunocompetent host (i.e. host immune cells) is necessary for the antitumor effects of GKT771, independently to the NOX1 expression level of cancer cells. *Firstly*, treatment of MC38 mouse colorectal tumor with GKT771 in immunocompromised mice does not result in antitumor effects in contrast to the antitumor effects obtained with GKT771 treatment in immunocompetent mice. *Secondly*, we used human colorectal cancer cells DLD1 that express high NOX1 mRNA level as a second cell line model, and LOVO which express low NOX1 mRNA level, as a third model in immunocompromised mice and show that GKT771 has no effect in any of tumors derived from these lines. *Thirdly*, we show that mice genetically deficient for NOX1 have impaired tumor growth.

We have observed that GKT771 treatment of B16F10 and MC38 cells in vitro results in decreased growth, thus confirming that GKT771 can have direct effects on tumor cells, but these effects were not dominant for the anti-tumor effect of GKT771 in vivo.

For this apparent discrepancy with previously published data we have no definitive explanation. However, there is a major difference in the experimental setting: the reported experiments were performed by constitutively silencing NOX1 expression in cancer cells while our experiments involved acute inhibition of NOX1 function. Silenced cells were selected and allowed to adapt to NOX1 deficiency in vitro before use, while in our experiments, cells were directly and acutely exposed to the inhibition in vivo; NOX1-silenced tumor cells were deficient from the time of injection in the mice and could have affected tumor take (a sensitive phase of tumorigenesis in transplantation models), while in our model, we treated tumors with GKT771 once they were already established (hence after succeeding tumor take). These data were further corroborated by effectively treating larger tumors with the inhibitor, a condition more relevant and similar to a clinical situation.

b. Absent functional NOX1 expression, the in vitro results presented in Supplemental Fig. 3E, are difficult to interpret. What is the mechanism by which GKT771 inhibits tumor cell or, for that matter, endothelial cell proliferation (Fig. 1J) in vitro? While some potential mechanisms of in vivo growth inhibition are addressed in the paper, there is essentially no such data provided for the in vitro studies-which has major impact for understanding the results presented.

Thank you for this important comment. We agree that decrease of cell numbers could be due to multiple reasons, in particular increased apoptosis, cell cycle alteration, cellular senescence, decreased cell adhesion and cell loss. To address this question, we performed additional experiments to determine the putative mechanism responsible for the decreased number of tumor cells treated with GKT771 in vitro. For endothelial cells, we already showed that GKT771 treatment decreases cell proliferation as measured by BrDU incorporation (Fig. 1G in the manuscript).

We performed a new flow cytometry cell cycle analysis study by EdU incorporation and propidium iodide staining and cell apoptosis measurement by Annexin V/propidium iodide staining in B16F10 exposed for 72 hours to GKT771, GKT831 and DMSO. Flow cytometry analysis revealed a decrease in cell cycle progression in cells treated with either GKT771 (75% in G0/G1, 6% in S phase and 18% in G2/M phase) and GKT831 (77% in G0/G1, 8.2% in S phase and 14.5% in G2/M phase) compared to DMSO-treated cells (63% in G0/G1, 8.5% in S phase and 28.5% in G2/M phase) (Fig.4A).

In contrast, neither GKT771 nor GKT831 could induce apoptosis in HUVEC or B16F10 cells (Fig.4B).

Figure 4: Cell cycle (A) and cell apoptosis (B) measurement by flow cytometry in B16F10.

By the present experiments, we confirmed that GKT771 decrease cell cycle progression in HUVEC and tumor cells. The inhibition of cell proliferation and cell cycle by GKT771 is interesting, also in view of what was discussed above (i.e. role of tumoral vs stomal NOX1) and will be further investigated in future experiments.

2. With respect to experimental methodology, the authors have chosen to treat tumors in vivo at a size (50 mm³) that is essentially unmeasurable. Put another way, the results demonstrate an adjuvant effect rather than a therapeutic effect on well-established tumors. This is a very important distinction in terms of how the data are to be interpreted; the paper would be much stronger if the effects of GKT771 on tumors that can be measured (≥ 200 mm³) were assessed, since this would have major implications regarding how one might think about the value of NOX1 inhibition in various clinical situations, and the potential therapeutic value of GKT771 in particular.

Thank you for your comment, which is relevant also in terms of the potential clinical use of the inhibitor. In our models, tumors at 50 mm³ are detectable but we also agree that by starting treatment at 50 mm³ we may be rather early relative to a potential clinical condition and may therefore lead to overestimating the potential effects on larger tumors. This protocol was originally chosen to allow for sufficient treatment time before tumors reached a volume of 1 cm³, which is the maximum tumor size allowed by our animal experimentation authorization in Switzerland, while on the one side to confirm successful tumor take upon tumor cell injection.

We nevertheless agree that the comment is relevant in order to mimic a clinically-relevant condition and test a true therapeutic effect of the drug. We therefore decided to perform new experiments by starting treatment at tumor size around 200 mm³, which can be considered more representative of tumors in clinical settings.

In this new protocol, we treated mice for only 4 days compared to the previous protocol in which mice were treated for 10 days. Under these new experimental conditions, tumor volume and tumor weight measurements at the end of the experiment, demonstrate that the 4 days GKT771 treatment inhibited tumor growth by about 50% for B16F10 and 35% for MC38 (Fig.4). These new results indicate that GKT771 is efficient in inhibiting growth of well-established tumors, over a relatively short period of time, as it is the case in clinical situations.

Figure 4: In vivo tumor size and ex vivo tumor mass monitoring of MC38 and B16F10 treated with GKT771 and vehicle at 200 mm³.

We have mentioned these results in the revised manuscript in the Fig.1L and Supplementary Fig.1I and in the text:

GKT771 inhibits tumor growth of MC38-derived colon carcinoma and B16F10 melanoma cancer in a clinically relevant model

Starting treatment at 50 mm³ tumor size may overestimate the potential effects on larger tumors. We therefore repeated experiments with GKT771 by starting treatment at tumor size of around 200 mm³, for both MC38 and B16F10 tumor models, which can be considered more representative of tumors in clinical settings. Mice were treated during 4 days compared to 10 days in the previous protocol. Under these new experimental conditions, the 4 days GKT771 treatments inhibited tumor growth (measured as tumor volume and tumor weight at the end of the experiment) by 35% for MC38 (Fig.1L and 1M) and 50% for B16F10 (Supplementary Fig.1I and 1J).

These new results indicate that GKT771 is efficient in inhibiting growth of well-established tumors, as it is the case in clinical situations.

3. What are the pharmacokinetics of GKT771 in vivo in these tumor-bearing models? Do the in vivo drug levels come anywhere near the micromolar levels needed for growth inhibition and anti-angiogenic effects in vitro?

Indeed, in vivo pharmacokinetics biodistribution of GKT771 is missing in this manuscript. This is primarily due to the fact that analytical methods for GKT771 detection in tissue is still in progress and not available yet. Different techniques are under development and validation to detect and quantify GKT771 in tissue. Among them, radiolabeling and mass spectrometry are the most promising emerging ones.

Preliminary experiments using radiolabeled GKT771 have been performed, in rats, with radiolabeled ¹⁴C-GKT771. Interestingly, after one oral gavage or intravenous injection, GKT771 is still detected after 24h in liver, lung, kidney, bladder, adrenal and thyroid gland.

4. Do the authors have data to address the mechanism by which GKT771 diminishes VEGF secretion in vitro? Is it simply a result of decreased tumor cell proliferation?

Indeed, the reviewer is right, the diminished VEGF secretion could be an indirect effect secondary to reduced cell proliferation and tumor mass, in opposition to a direct effect on VEGF mRNA expression or protein synthesis. A third possibility is an effect on stromal cells, particularly myeloid-derived inflammatory cells, known to express and secrete VEGF.

To address this question, we performed VEGF mRNA expression analysis in the two mouse B16F10 and MC38 cancer cells treated in vitro with GKT771 and GKT831 compounds (or vehicle), and in tumors derived from the same cell lines in vivo, treated with GKT771 or vehicle.

VEGF mRNA expression was not suppressed in both cell lines by GKT771 or GKT831 treatment in vitro during 24 and 48h (rather GKT771 treatment tended to induce a VEGF upregulation) (Fig.5A).

When we analyzed VEGF mRNA expression in lysates of MC38 and B16F10 tumor treated in vivo with either GKT771 or vehicle, we observed a trend toward a decrease in VEGF mRNA levels in GKT771-treated mice, though differences we are not statistically significant (Fig. 5B).

Figure 5: A. VEGF mRNA relative expression in MC38 and B16F10 cell lines treated with GKT771, GKT831 or DMSO in vitro during 24 and 48h. B. VEGF mRNA relative expression in MC38 and B16F10 tumors treated in vivo with GKT771 or vehicle.

From these experiments, we can conclude that GKT771 treatment does not directly inhibit VEGF expression. The decreased VEGF secretion observed upon treatment may therefore be indirect due to reduced cell proliferation and reduced tumor growth rather than to a direct effect on VEGF synthesis.

We have mentioned these results in the revised manuscript in the text:

However, analysis of VEGF-A mRNA expression in cancer cells treated in vitro with GKT771 or GKT831 showed no decrease in expression. In contrast, VEGF-A mRNA measurement in tumors treated in vivo

with GKT771 showed a trend towards decreased levels (data not show). This suggests that the decrease of VEGF-A in supernatant and mouse plasma is related to a decreased cell proliferation in vitro and a decreased tumor growth in vivo.

5. One explanation for the synergistic effects of GKT771 and a PD-1 blocker is that the small molecule might enhance the mutational burden (and hence antigen presentation) of the tumor cells (as has been shown for treatment with temozolamide). Do the authors have any information on potential non-NOX1 related effects of GKT771 that might be consistent with this current hypothesis that underlies much of what is known regarding the mechanism of action of PD-1 antibodies?

Temozolomide is indeed known to be mutagenic in vitro and in vivo, through its alkylating activity, particularly in MGMT-low cells, and this translates into an increase in antigenicity of a treated tumor. Based on the chemical characteristics and on its mode of action, we have no reasons to believe that GKT771 itself is mutagenic, thereby making treated cells more immunogenic. As GKT771 decreases ROS production (known to be mutagenic) one would rather expect a decreased mutagenic rate.

6. As noted above, it is difficult to conclude that the intrinsic NOX1 levels of the tumor are unrelated to the apparent effects on the microenvironment when essentially all of the tumor cell lines studied either lack functional NOX1 or are likely to do so (Page 13); similarly, since tumor growth delay has been observed secondary to NOX1 knockdown in immunocompromised mice previously, it is quite difficult to conclude that antitumor effects of NOX1 inhibition require an intact immune system.

Rather, it would appear that an intact innate immune system may play a role in NOX1-related tumor cell growth. In particular, the authors should be aware that the partial growth delay observed for GKT771 treatment in both the B16 and MC38 systems was modest (tumors continued to grow over the 15 days of observation). Since larger tumors were not used, it is unclear whether the agent would cause tumor regressions per se.

As described in the mRNA expression experiments above and in the figure included in the original manuscript, the human and mouse cell lines that we used in the experiment express NOX1, albeit at different levels (MC38 > B16F10 and DLD1 > LoVo). We also performed experiments with MC38 colorectal cancer cell lines in both immunodeficient and immunocompetent mice and in syngeneic NOX1 KO mice. The lack of effects of GKT771 in both immunodeficient mice and NOX1 KO mice strongly indicates that host NOX1 and immune cells are needed for the anti-tumor effects of GKT771.

GKT771 has effects on cell proliferation in vitro, and we cannot exclude that it may also have direct effects on cancer cell proliferation in vivo. But these cell autonomous effects on tumor cells in vivo are not sufficient to reduce tumor growth in the absence of immune cells or in absence of NOX1 expression in the host.

Concerning the experiments done on small tumors, as shown above, we repeated treatment experiments using larger tumors (200mm³). Indeed, even with larger tumors we observed tumor growth reduction too. However, treatment could be applied for just 4 days due to the maximal tumor size allowed by the animal authorization protocol in Switzerland and in these experiments, tumor reduction

was around 35-50% compared to the 85% reduction when treatment was started when tumors were only 50mm³ in volume. Thus, we consider these effects important and highly relevant to clinical conditions.

Taken together, these results demonstrate that the anti-tumor effect observed by pharmacological inhibition of NOX1 with GKT771 requires that an intact immune system is present with host NOX1 expression. Considering the direct effects observed in vitro, it is possible that GKT771 may also have direct anti-tumor effects in vivo, though this is not what we observed in our experiments in immunosuppressed or NOX1 KO mice. The reason for this apparent discrepancy with previous publications is not clear at this point (See also Reviewer 2, query 1 above). We have highlighted this issue in the discussion.

7. The Methods state that experiments were also performed with SW480 and HCT-116 cell lines; what were those results?

We agree with this comment. Indeed, we tested NOXs mRNA expression on these cell lines but as no further experiments were done with these lines we removed to mention them in the materials and methods section.

May 17, 2019

RE: Life Science Alliance Manuscript #LSA-2018-00265-TR

Dr. Jimmy Stalin
Faculty of Science and Medicine University of Fribourg
Department of Oncology/Microbiology/Immunology (OMI)
Ch. du Musée 18, PER17
Fribourg, Canton de Fribourg 1700
Switzerland

Dear Dr. Stalin,

Thank you for submitting your revised manuscript entitled "Inhibition of host NOX1 blocks tumor growth and enhances checkpoint inhibitor-based immunotherapy". We have now assessed your work and the point-by-point response provided to the concerns initially raised. We appreciate the introduced changes and think that they address the reviewer concerns well. However, a few aspects need to get better incorporated in the main manuscript prior to acceptance here:

- it is important to include the Nox1-specificity assay (currently in pbp response to rev#1's concern) in the manuscript. If this is not possible, the related work you are referring to and that includes the assay should get uploaded to BioRxiv to allow citation of the preprint article as support for specificity
- please state in the manuscript that in vivo pharmacokinetics (point 3 of rev#2) will have to be tested in the future
- please mention in the text that potentially a post-transcriptional effect leads to the observed reduced VEGF-A secretion (rev#2, point 4)

Please also address the following editorial points:

- please indicate in the manuscript text file the corresponding author
- the corresponding author profile in our submission system needs to get linked to an ORCID iD, please initiate this (emails with instructions have been sent to authors)
- please add the number of biological and technical replicates used in the qPCR analysis
- please add a scale bar to Fig1E

A. FINAL FILES:

B. MANUSCRIPT ORGANIZATION AND FORMATTING:

Thank you for your attention to these final processing requirements.

Sincerely,

Andrea Leibfried, PhD

Executive Editor
Life Science Alliance
Meyrhofstr. 1
69117 Heidelberg, Germany
t +49 6221 8891 502
e a.leibfried@life-science-alliance.org
www.life-science-alliance.org

1/- It is important to include the Nox1-specificity assay (currently in pbp response to rev#1's concern) in the manuscript. If this is not possible, the related work you are referring to and that includes the assay should get uploaded to BioRxiv to allow citation of the preprint article as support for specificity.

We agree to provide more information about the characterization of the compounds. To this end, this sentence was added in the manuscript in the Results section:

A potent and highly selective NOX1 inhibitor (GKT771) has been developed by Genkyotex. As shown in Supplementary Fig. 1A. 1, GKT771 inhibits NOX1 with an inhibitory constant K_i of 60 ± 6 nM. It is highly selective over NOX4 ($K_i = 4000 \pm 400$ nM) and is inactive against all other NOX isoforms. Moreover, GKT771 shows to be inactive on all counterscreen assays, including xanthine oxidase, glucose oxidase and scavenging assays (data not shown). Together with the availability of NOX1-deficient mice, GKT771 provides a unique opportunity to investigate selective NOX1 inhibition as a potential anticancer strategy.

In addition, the paragraph about characterization of the compounds has been modified and a reference added in the Materials and Methods section:

Inhibitors

GKT771 and GKT831 NOX inhibitors were developed by Genkyotex S.A. using recombinant cells transfected with the NOX1 to NOX5 isoforms. Activity of compounds against the different NOX isoforms was measured as previously described [50]. Briefly, membranes prepared from the different recombinant cell lines were incubated in PBS with Amplex Red, horseradish peroxidase (HRP), and appropriate cofactors. ROS production was induced by the addition of NADPH. Inhibitory activity of compounds was measured in the presence of increasing concentrations ranging from 1 nM to 100 μ M. After 20 min of incubation at 37°C. ROS levels were measured using a BMG Labtech microplate reader.

Finally, a description of the new results was added in the Supplementary figure legends section and associated panel included in the Supplementary Figure 1:

Inhibition of NOX dependent ROS production on membranes. A. Concentration-response curves of GKT771 on membranes prepared from cells specifically over-expressing the indicated human NOX isoform. Results shown are from one experiment performed in triplicate, representative of at least three independently performed experiments.

2/- Please state in the manuscript that in vivo pharmacokinetics (point 3 of rev#2) will have to be tested in the future.

This interesting aspect for the potent biodistribution and effects of GKT771 is now discussed in the discussion part of the manuscript (Page 13-14). We have added the following paragraph:

We will address this issue in future experiments for example with pharmacokinetics studies in tumor models. Indeed, it will be of interest to determine the amount of GKT771 into tumors and healthy tissues/organs to assess its biodistribution and tumor penetration.

3/- Please mention in the text that potentially a post-transcriptional effect leads to the observed reduced VEGF-A secretion (rev#2, point 4).

Thanks for your comment, it is true that we did not discuss this aspect in the manuscript. We add a paragraph in the discussion part of the manuscript (Page 14). We have add the following paragraph:

NOX1 inhibition in tumor cells in vitro decreased the production of VEGF-A and PLGF, two important angiogenic factors associated with tumor growth and progression [41-42]. We did not observe any inhibitory effect of GKT771 on VEGF-A mRNA expression levels suggesting that the decreased VEGF-A secretion by cells in vitro due to NOX1 inhibition may involve pro-translational regulation, such as protein maturation, transport and secretion, or could be due to a decrease in cell number.

4/- Please also address the following editorial points:

Please indicate in the manuscript text file the corresponding author.

The names of the corresponding authors have been added to the manuscript text files. We have added the following paragraph (page 1):

Corresponding authors and ORCID numbers:

Corresponding authors and ORCID numbers:

Jimmy Stalin (0000-0001-8092-7057) jimmy.stalin@unifr.ch

Curzio Ruegg (0000-0001-9137-7695) curzio.ruegg@unifr.ch

Beat A. Imhof (0000-0002-6446-3990) beat.imhof@unige.ch

The corresponding author profile in our submission system needs to get linked to an ORCID iD, please initiate this (emails with instructions have been sent to authors).

We put the ORCID identification number of the three corresponding authors in the manuscript and the submission system as asked.

Please add the number of biological and technical replicates used in the qPCR analysis.

Thank you for this comment. We added the number of technical and biological replicates during qPCR experiments in the appropriate paragraph in the materials and methods section in the quantitative polymerase chain reaction by adding this sentence page 18:

The qRT-PCR were routinely performed with two technical replicates and three biological replicates.

Please add a scale bar to Fig1E.

Thanks for the comment about missing scale bar in Fig1E. We added scale bar (80 pixels, white) in the 3 aortic ring pictures of Fig1E using Image J software. Images were also enlarged. A sentence was added in the figure legends section for scale bar size of Fig1E:

Scale bars were added in the aortic ring images by using Image J software. White scale bar have a size of 1000 μm .

June 13, 2019

RE: Life Science Alliance Manuscript #LSA-2018-00265-TRR

Dr. Jimmy Stalin
Faculty of Science and Medicine University of Fribourg
Department of Oncology/Microbiology/Immunology (OMI)
Ch. du Musée 18, PER17
Fribourg, Canton de Fribourg 1700
Switzerland

Dear Dr. Stalin,

Thank you for submitting your Research Article entitled "Inhibition of host NOX1 blocks tumor growth and enhances checkpoint inhibitor-based immunotherapy". I appreciate the introduced changes and it is a pleasure to let you know that your manuscript is now accepted for publication in Life Science Alliance. Congratulations on this interesting work.

DISTRIBUTION OF MATERIALS:

Again, congratulations on a very nice paper. I hope you found the review process to be constructive and are pleased with how the manuscript was handled editorially. We look forward to future exciting submissions from your lab.

Sincerely,
